# From Observations to Events: Event-Aware World Model for Reinforcement Learning

**Zhao-Han Peng**[1]**, Shaohui Li**[2]**, Zhi Li**[1]**, Shulan Ruan**[1]**, Yu Liu**[3]*****, You He**[3]

[1]Shenzhen International Graduate School, Tsinghua University
[2]College of Information Science and Electronic Engineering, Zhejiang University
[3]Department of Electronic Engineering, Tsinghua University

## Abstract

While model-based reinforcement learning (MBRL) improves sample efficiency by learning world models from raw observations, existing methods struggle to generalize across structurally similar scenes and remain vulnerable to spurious variations such as textures or color shifts. From a cognitive science perspective, humans segment continuous sensory streams into discrete events and rely on these key events for decision-making. Motivated by this principle, we propose the Event-Aware World Model (EAWM), a general framework that learns event-aware representations to streamline policy learning without requiring handcrafted labels. EAWM employs an automated event generator to derive events from raw observations and introduces a Generic Event Segmentor (GES) to identify event boundaries, which mark the start and end time of event segments. Through event prediction, the representation space is shaped to capture meaningful spatio-temporal transitions. Beyond this, we present a unified formulation of seemingly distinct world model architectures and show the broad applicability of our methods. Experiments on Atari 100K, Craftax 1M, and DeepMind Control 500K, DMC-GB2 500K demonstrate that EAWM consistently boosts the performance of strong MBRL baselines by 10%–45%, setting new state-of-the-art results across benchmarks. Our code is released at https://github.com/MarquisDarwin/EAWM.

## 1 Introduction

Historically, policy learning in complex environments within model-free reinforcement learning requires millions of interactions with environments (Jaderberg et al., 2017), which limits its real-world applications. MBRL algorithms utilize world models to capture the dynamics of the environment (Ha & Schmidhuber, 2018), learn representations from high-dimensional observations (Ebert et al., 2018; Hafner et al., 2019b; Zhang et al., 2019), and imagine future frames (Hafner et al., 2019a; Kaiser et al., 2020). Once the world model is established, the policy can be optimized on imagined trajectories, reducing reliance on real-world interaction and thereby improving sample efficiency.

Existing MBRL approaches focus on learning world models that predict future states through self-supervised learning in the observation space. As a notable example, DreamerV3 (Hafner et al., 2025) estimates distributions over prior states (before observations) and posterior states (after observations) to handle stochasticity. Transformer world models like TWISTER (Burchi & Timofte, 2025), RetNet-based (Sun et al., 2023) models like Simulus (Cohen et al., 2025), and diffusion models (Alonso et al., 2024) were exploited to generate precise future observations. However, the training objectives in the observation space result in inaccuracy of long-horizon prediction (Li et al., 2025) and redundant information for policy learning. Moreover, predicting raw observations in stochastic environments is inherently ill-posed: for example, one cannot predict the outcome of a coin flip before it occurs.

In contrast, biological systems treat events as fundamental units of perception and action (Wahlheim & Zacks, 2025). Inherent advantages of processing events include lower power consumption and lower latency, and theoretical optimality for decision-making (Butz et al., 2021). Recent advances (González-Rueda et al., 2024; Lu et al., 2024) in neurobiology have revealed that the superior colliculus (SC)

---

*Corresponding to Yu Liu (`liuyu_thu@mail.tsinghua.edu.cn`).

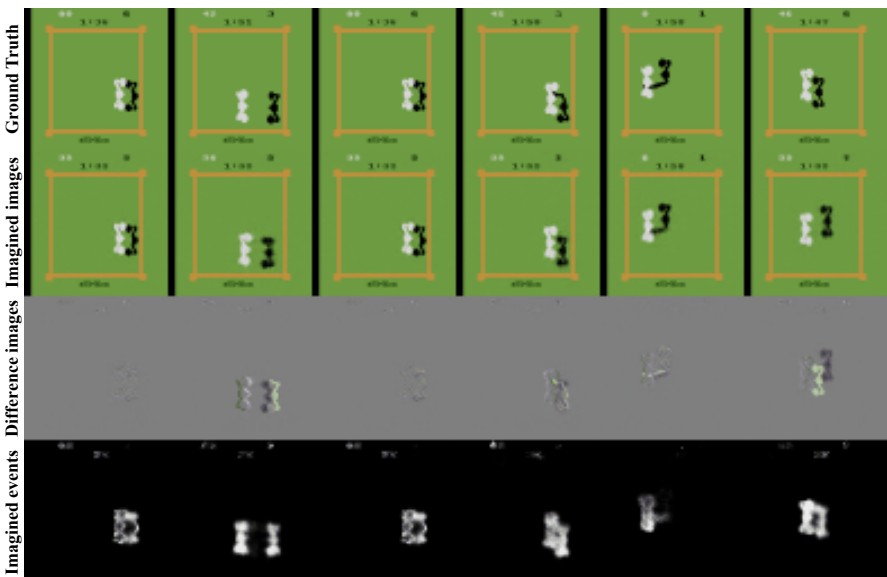

Figure 1. Imagined frames and event predictions under the framework of EAWM at imagination step 9. The first two rows show ground-truth and imagined images, while the third row highlights their pixel-wise differences. Each column corresponds to a distinct trajectory. Notably, although the imagined frames may deviate from the ground truth in object positions, the event predictor consistently localizes spatial boundaries accurately.

contains neurons that respond exclusively to changes in visual scenes (e.g., looming predators or moving targets), enabling rapid behavioral responses. In addition, SC neurons align sensory inputs with motor commands in vector space, not just spatial coordinates. This allows real-time interception of moving objects by mapping dynamic sensory features like velocity to action trajectories (Hunt et al., 2025). These findings suggest that animals often respond directly to dynamic events rather than relying solely on static observations. Consider, for instance, a scenario where a large, unfamiliar creature rapidly approaches in a forest: a human would instinctively flee without waiting to fully identify the creature. In contrast, current world models, driven primarily by observation reconstruction, are prone to failure in such scenarios, particularly when faced with novel objects or dynamics absent from the training data. They put too much emphasis on observation predictions, leading to poor generalization and ineffective policy learning. Nevertheless, humans' brains only make predictions of events rather than predictions of future observations (Ben-Yakov & Henson, 2018). This highlights the need for building world models that capture kinetic features for robust agent control—a direction that has been largely neglected in prior MBRL research.

To that end, we present a unified framework called Event-Aware World Model (EAWM), which introduces a generally applicable method that learns compact kinetic features from raw observations in the environment. Yet how SC neurons process kinetic features remains to be explored. We argue that event prediction is an effective way to construct an event-aware agent within a computational framework, instead of building a complicated system to filter out noise and take events as inputs (Gehrig & Scaramuzza, 2024). Because event prediction intrinsically constrains the representation space to meaningful spatio-temporal transitions through information bottleneck optimization. As illustrated in Figure 1, we find that event prediction is inherently less complex than observation prediction, since it abstracts away redundant information or low-level variation. We conduct experiments and demonstrate the strong performance and adaptability of EAWM for diverse scenarios, as shown in Figure 2. We observe that EAWM enhances the performance of base world models by a large margin in tasks requiring event-aware reactions, for example, with improvements of 55% in *Breakout* and 115% in *Acrobot Swingup*. The main contributions of this work are summarized as follows:

- To the best of our knowledge, this paper, for the first time, systematically analyzes the advantages of kinetic features for policy learning and introduces a general framework to learn concise latent representations of events without requiring any manual labels.

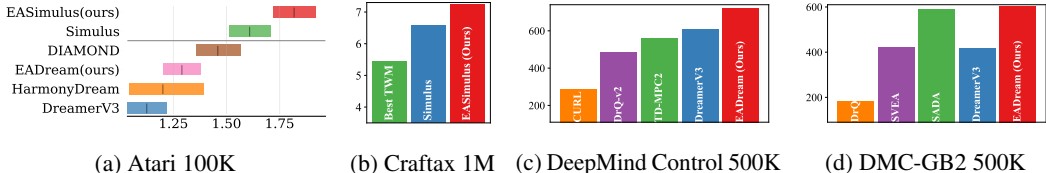

(a) Atari 100K     (b) Craftax 1M   (c) DeepMind Control 500K    (d) DMC-GB2 500K

Figure 2. **Overview**. EAWM surpasses existing model-free and model-based RL across 55 test tasks, encompassing both continuous and discrete control, as well as multi-modal observations. (a) Mean human-normalized scores and the $95\%$ stratified bootstrap confidence intervals (Agarwal et al., 2021) on the 26 tasks of Atari 100K. (b) Percentage of scores against the maximum score in Craftax. (c) Mean returns on 10 challenging tasks from DeepMind Control Suite. (d) Mean returns over 6 tasks on 3 test environments from DMC-GB2.

- We design a generic event segmentor (GES) to identify event boundaries, which enable robust representation learning responsive to the critical events for multimodal observations.
- We show the broad applicability of our methods through a unified formulation of seemingly distinct world model architectures, as demonstrated by our implementations of EADream and EASimulus.
- With minimal tuning, we show that our methods improve the baseline world models by 10%–45%, setting new records across diverse established MBRL benchmarks.

## 2 BACKGROUND

### 2.1 REINFORCEMENT LEARNING

Reinforcement learning in the real world can be formalized as a Partially Observable Markov Decision Process (POMDP) (Kaelbling et al., 1998) with observations $\mathbf{o}_t \in \Omega$, actions $\mathbf{a}_t \in \mathcal{A}$, rewards $r_t \in \mathbb{R}$, states $\mathbf{s}_t \in \mathcal{S}$, and a discount factor $\gamma \in (0, 1]$. An agent takes an action according to the policy $\pi(\cdot|\mathbf{o}_{\leq t}, \mathbf{a}_{<t})$, which is a mapping from the history of past observations and actions to a probability distribution on actions to take. The object is to learn a policy $\pi$ that maximizes the expected value of accumulated discounted reward $\mathbb{E}_\pi[\sum_{t=0}^\infty \gamma^t r_t | \mathbf{s}_0 = \mathbf{s}]$.

### 2.2 EVENT DEFINITION

Events are fundamentally defined as a segment of time at a given location that is perceived by an observer to possess a distinct commencement and termination (Zacks & Tversky, 2001). Extensive research has established that events are encoded, perceived, and cognitively processed in the human brain (Wahlheim & Zacks, 2025).

**Visual Inputs**. Within visual sensing, an event constitutes a localized variation in the logarithm of brightness (Gallego et al., 2020). Herein, the brightness $I_t(x_i, y_i)$ is proportional to the irradiance hitting the pixel $(x_i, y_i)$ at the 2D camera plane at temporal point $t$. Let $L_t(x_i, y_i) \doteq \log I_t(x_i, y_i)$. An event $e_{t,i}^{\text{vis}} = (x_i, y_i, t, p_i)$ is triggered at pixel $(x_i, y_i)$ when the magnitude of change attains a predetermined temporal contrast threshold (Chakravarthi et al., 2024), denoted $C_I$:

$$p_i = \begin{cases} +1, & \text{if } L_t(x_i, y_i) - L_{t-1}(x_i, y_i) > C_I \\ -1, & \text{if } L_t(x_i, y_i) - L_{t-1}(x_i, y_i) < -C_I \\ 0, & \text{otherwise} \end{cases} \quad (1)$$

$p_i(\neq 0)$ denotes the existence of events, and the sign of the brightness change denotes the direction of the event. Consequently, events can be derived algorithmically via established computational frameworks.

**Ordinal Data vs. Nominal Data**. Regarding ordinal data possessing inherent ordering characteristics (e.g., proprioceptive vectors), the event $e_{t,i}^{\text{ord}} = (i, t, p_i)$ is defined as follows:

$$p_i = \begin{cases} +1, & \text{if } (o_t(i) - o_{t-1}(i))/\text{Range}(o_i) > C_o \\ -1, & \text{if } (o_t(i) - o_{t-1}(i))/\text{Range}(o_i) < -C_o \\ 0, & \text{otherwise} \end{cases} \quad (2)$$

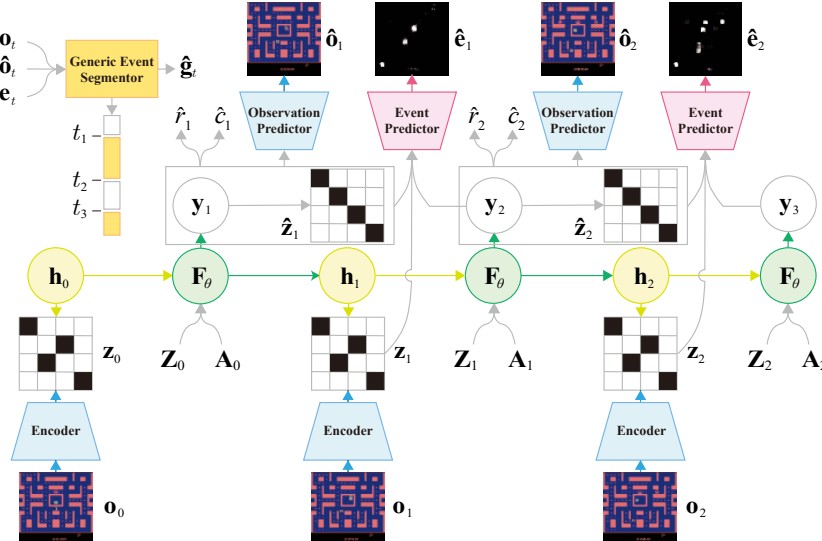

Figure 3. EAWM architecture that predicts the next observations and events. Given the length of the trajectory segment $k$, the sequence model outputs $\mathbf{y}_t$, which summarizes the embeddings $\mathbf{Z}_{t-1} = [\mathbf{z}_{t-k}, ..., \mathbf{z}_{t-1}]$ and $\mathbf{A}_{t-1} = [\mathbf{a}_{t-k}, ..., \mathbf{a}_{t-1}]$. The observation predictor predicts the next observation $\hat{\mathbf{o}}_t$ via the outputs of the sequence model $\mathbf{y}_t$ and the outputs of the dynamics predictor $\hat{\mathbf{z}}_t$. The representation model combines observation encodings with hidden states $\mathbf{h}_t$ to obtain embeddings $\mathbf{z}_t$. The generic event segmentor identifies event boundaries, the starting and ending points of meaningful segments of the observation stream.

where $o_t(i)$ is the value of the $i$-th ordinal data at time $t$ and $\text{Range}(o(i))$ is the range of ordinal data $o(i)$. From the above equations, event generation becomes straightforward once the threshold $C_o$ is specified (Hunt et al., 2025). For nominal data comprising mutually exclusive categories, we define an event $e_{t,i}^{\text{nom}}$ as a change in the category of the corresponding observation. As listed in Table 12, the threshold for each modality is fixed across benchmarks.

## 3 METHODS

### 3.1 EAWM FRAMEWORK

**Components** Given trajectory segments of observations $\mathbf{O}_t = [\mathbf{o}_{t-k+1}, ..., \mathbf{o}_t]$, segments of actions $\mathbf{A}_t = [\mathbf{a}_{t-k+1}, ..., \mathbf{a}_t]$, rewards $r_t$, continuation flags of episodes $c_t$, and events $\mathbf{e}_t$, world models can be trained in a self-supervised manner. The general formulation of each component of EAWM is outlined below:

$$
\begin{cases}
\text{WM} \begin{cases}
\text{Sequence Model:} & \mathbf{h}_t, \mathbf{y}_t = \mathbf{F}_\theta(\mathbf{h}_{t-1}, \mathbf{Z}_{t-1}, \mathbf{A}_{t-1}) \\
\text{Representation Model:} & \mathbf{z}_t \sim q_\theta(\mathbf{z}_t \,|\, \mathbf{o}_t, \mathbf{h}_t) \\
\text{Dynamics Predictor:} & \hat{\mathbf{z}}_t \sim p_\theta(\hat{\mathbf{z}}_t \,|\, \mathbf{y}_t) \\
\text{Reward Predictor:} & \hat{r}_t \sim p_\theta(\hat{r}_t | \mathbf{y}_t, \mathbf{z}_t) \\
\text{Continuation Predictor:} & \hat{c}_t \sim p_\theta(\hat{c}_t | \mathbf{y}_t, \mathbf{z}_t) \\
\text{Observation Predictor:} & \hat{\mathbf{o}}_t \sim p_\theta(\hat{\mathbf{o}}_t | \mathbf{y}_t, \hat{\mathbf{z}}_t)
\end{cases} \\
\text{EA} \begin{cases}
\text{Event Predictor:} & \hat{\mathbf{e}}_t \sim p_\theta(\hat{\mathbf{e}}_t | \mathbf{y}_t, \mathbf{y}_{t+1}, \hat{\mathbf{z}}_t, \mathbf{z}_t) \\
\text{Generic Event Segmentor:} & \hat{\mathbf{g}}_t = g_\theta(\mathbf{o}_t, \hat{\mathbf{o}}_t, \mathbf{e}_t)
\end{cases}
\end{cases}
\tag{3}
$$

where $\mathbf{h}_t$ is the hidden state, $\mathbf{y}_t$ the output of the sequence model, and $\mathbf{z}_t$ the embeddings of observations. The tilde here indicates random variables sampled from their corresponding distribution. The WM part of Equation 3 is a general summary for modern world models, and almost all of them (Seo et al., 2023; Robine et al., 2023; Zhang et al., 2024b; Burchi & Timofte, 2025; Hafner et al., 2025) have included these five components shown above since they do exactly what needs to

be done in MBRL. Notably, the EA part can be predicted with the output of the WM part. Hence, our event-aware representation learning in Section 3.3 can be incorporated with these world models without theoretical difficulty. That is to say, our proposed framework is generally applicable, even to model-free RL.

**Behavior Learning** Our agent learns behaviors entirely from imagined trajectories generated by the world model, without requiring additional interactions with the external environment. If world models generated precise predictions of future observations and events in the environment, they would inherently learn robust and compact representations of history. We define the agent's state $\mathbf{s}_t$ as the integration of the sequence model outputs $\mathbf{y}_t$ with the observation embeddings $\mathbf{z}_t$. This latent state serves as the input to standard reinforcement learning algorithms such as REINFORCE (Sutton et al., 1999), enabling policy optimization within the imagined environment. Importantly, the prediction of events or observations has no direct influence on policy training; that is to say, the computational overhead of the observation predictor and the event predictor can be avoided during policy learning.

## 3.2 MULTI-MODAL EVENTS

**Automated Event Generator** To capture kinetic features explicitly, we introduce an automated process for event generation, thereby eliminating the need for manual labeling. As aforementioned, the notion of an event depends on the modality of the observations. For visual inputs, instead of using the primitive definition of events, we employ Adaptive Gaussian Mixture Models (AGMMs) (Zivkovic & Van Der Heijden, 2006) to reduce false alarms from noise or gradual changes in brightness. That is, we treat the event as a statistically significant deviation from a learned multi-modal distribution of each pixel. The model maintains a mixture of $K$ Gaussian components for each pixel:

$$p(L_t(x_i, y_i)) = \sum_{k=1}^{K} w_{k,t,i} \mathcal{N}\big(L_t(x_i, y_i); \mu_{k,t,i}, \Sigma_{k,t,i}\big),$$

where $w_{k,t,i} \geq 0$ are the mixture weights with $\sum_{k=1}^{K} w_{k,t,i} = 1$, $\mu_{k,t,i}$ are the component means, and $\Sigma_{k,t,i}$ are the covariance matrices. We compute the squared Mahalanobis distance $D_{k,i}$ between the next observation $L_{t+1}(x_i, y_i)$ and the component means $\mu_{k,t,i}$ for each component $k$:

$$D_{k,i} = (L_{t+1}(x_i, y_i) - \mu_{k,t,i})^\top \Sigma_{k,t,i}^{-1} (L_{t+1}(x_i, y_i) - \mu_{k,t,i}).$$

If $D_{k,i}$ is larger than the threshold $C_I$, we generate an event $e_{t+1,i}^{\text{vis}} = (x_i, y_i, t+1, p_i)$ on that pixel and a new component will be generated; otherwise, $L_{t+1}(x_i, y_i)$ matches a Gaussian component, say $k$, and the weight of component $k$ will be increased. We also generate an event $e_{t+1,i}^{\text{vis}}$ when the matched component exhibits low weight in comparison to other components. To summarize, events are generated at spatio-temporal locations where the model is either surprised (large $D_{k,i}$) or uncertain (low weight), which aligns well with event-based systems (Muir & Sheik, 2025) that concentrate computation on informative changes rather than on every frame and every pixel. As a result, the event stream is sparse yet information-rich, and less sensitive to irrelevant noise and slow illumination changes than the raw observations.

As described in Section 2.2, for ordinal data such as joint angles and velocity, an event $e_{t,i}^{\text{ord}} = (i, t, p_i)$ will be generated whenever the magnitude of change reaches the threshold $C_o$. For nominal data like tokenized inputs, an event $e_{t,i}^{\text{nom}}$ occurs whenever the type of information changes. This modular event-generation process enables our model to flexibly accommodate diverse observation modalities. Full implementation details and configurations are provided in Appendix D.

**Notations** For multi-modal observations $\mathbf{o}_t = \{\mathbf{o}_t^{(m)}\}_{m=1}^M$, where $\mathbf{o}_t^{(m)}$ denotes the features of the $m$-th modality and $M$ represents the number of modalities, embeddings $\mathbf{z}_t = \{\mathbf{z}_t^{(m)}\}$ can be directly obtained from $m$-th modality of observations and $\mathbf{y}_t$, that is, $\mathbf{z}_t^{(m)} \sim q_\theta(\mathbf{z}_t^{(m)}|\mathbf{o}_t^{(m)}, \mathbf{y}_t)$. Correspondingly, the distribution of the observations of the $m$-th modality is $p_\theta^{(m)}(\hat{\mathbf{o}}_t|\mathbf{y}_t, \hat{\mathbf{z}}_t^{(m)})$ and the distribution of the events of the $m$-th modality is $p_\theta^{(m)}(\hat{\mathbf{e}}_t^{(m)}|\mathbf{y}_t, \mathbf{y}_{t+1}, \hat{\mathbf{z}}_t^{(m)}, \mathbf{z}_t^{(m)})$.

## 3.3 EVENT-AWARE REPRESENTATION LEARNING

**Event Predictor** To learn event-aware representations for world models, we design the event predictor to capture events in the observations. For each modality of events $\hat{\mathbf{e}}_t^{(m)}$, we design a

separate decoder network to estimate $p^{(m)}(\hat{e}_{t,i}^{(m)}|\text{sg}(\mathbf{y}_t), \mathbf{y}_{t+1}, \hat{\mathbf{z}}_t^{(m)}, \mathbf{z}_t^{(m)}) \in [0,1]$, the probability of the class of every event $e_{t,i}^{(m)}$. To avoid spikes in the loss and encourage representations to be more meaningful and predictable, the stop gradient operator $\text{sg}(\cdot)$ is used to prevent the gradients of targets from being backpropagated.

As aforementioned, we categorize the observations into three types, *i.e.*, visual inputs $\mathcal{D}_I$, ordinal data $\mathcal{D}_o$, and nominal data $\mathcal{D}_n$. If the modality of observations falls into ordinal data, the cross-entropy loss is applied. If the modality of observations falls into nominal data like visual inputs, the focal loss (Lin et al., 2017) is applied:

$$\mathcal{L}_{\text{e}}^{(m)}(\theta) = \begin{cases} \sum_{i=1}^{N^m} \text{CrossEntropy}(p_i^{(m)}, \hat{p}_i^{(m)}) & \text{if } m \in \mathcal{D}_o \\ \sum_{i=1}^{N^m} \text{Focal}(p_i^{(m)}, \hat{p}_i^{(m)}) & \text{if } m \in \mathcal{D}_I \cup \mathcal{D}_n \end{cases}, \tag{4}$$

where $N^m$ denotes the dimensionality (size) of the observation for the $m$-th modality. The total loss of event prediction is the weighted sum of the prediction loss of each modality:

$$\mathcal{L}_{\text{e}}'(\theta) \doteq \sum_{m=1}^{M} \beta_{\text{e}}^{(m)} \mathcal{L}_{\text{e}}^{(m)}(\theta), \tag{5}$$

where the constant $\beta_{\text{e}}^{(m)}$ controls the scale of the event prediction loss of the $m$-th modality.

**Generic Event Segmentor** Excessive focus on transient fluctuations can divert attention from critical events, just as humans struggle to read when there is a sudden jolt (Katsuki & Constantinidis, 2012). The start or end time of a meaningful slice of an observation stream, such as the jolt, is referred to as the event boundary (Zacks & Swallow, 2007). Human comprehension systems tend to form future prediction within event boundaries, whereas humans' ability of prediction and memory decreases at the event boundaries (Radvansky & Zacks, 2017). Motivated by this, we develop a generic event segmentor (GES) that automatically detects event boundaries. For each modality $m$, we notice that the number of events $\alpha_t^{(m)} = \sum_{i=1}^{N^m} \frac{1}{N^m} \mathbb{I}(e_{t,i}^{(m)} \text{ occurs})$ can serve as an intuitive indicator of event boundaries. To keep the method computationally efficient, we do not introduce any additional trainable parameters into the GES. Instead, we implement the GES as a deterministic function of the events $\mathbf{e}_t^{(m)}$:

$$g_\theta(\mathbf{o}_t^{(m)}, \hat{\mathbf{o}}_t^{(m)}, \mathbf{e}_t^{(m)}) \doteq g(\alpha_t^{(m)}, \alpha_{\text{thr}}^{(m)}), \tag{6}$$

where $\alpha_{\text{thr}}^{(m)}$ denotes the threshold of the percentage of events that determines whether the agent should attend to events in the $m$-th modality. If $g(\alpha_t^{(m)}, \alpha_{\text{thr}}^{(m)}) = 0$, we say that an event boundary is detected. In this case, event prediction is devoid of meaning and should be suppressed. Therefore, Equation 4 with GES turns into:

$$\mathcal{L}_{\text{e}}(\theta) = \sum_{m=1}^{M} \beta_{\text{e}}^{(m)} g(\alpha_t^{(m)}, \alpha_{\text{thr}}^{(m)}) \mathcal{L}_{\text{e}}^{(m)}(\theta). \tag{7}$$

Furthermore, world models should prioritize dynamic events over static observations so as to improve accuracy on informative parts of the observations, rather than uniformly over all observations. However, when GES detects an event boundary where the priority of dynamic events is not suitable, the world model should reallocate attention from events to raw observations and focus on modeling raw observations. We implement this via an event-aware observation loss:

$$\mathcal{L}_{\text{o}}(\theta) \doteq \mathcal{L}_{\text{o}}'(\mathbf{o}_t, \hat{\mathbf{o}}_t) + \sum_{m=1}^{M} \sum_{i=1}^{N^m} \omega g(\alpha_t^{(m)}, \alpha_{\text{thr}}^{(m)}) \left[ \mathbb{I}(e_{t,i}^{(m)} \text{ occurs}) - 1 \right] \mathcal{L}_{\text{o}}'(o_{t,i}, \hat{o}_{t,i}), \tag{8}$$

where $\omega \in [0,1]$ is the event-aware weight to balance attention of the overall observations and the part of the observations where events take place. $\mathcal{L}_{\text{o}}'$ denotes the loss function of observations given by base world models by default if it exists. Otherwise, the mean square error of observations is calculated. Overall, the total loss of the EAWM is

$$\mathcal{L}_{(}\theta) \doteq \mathcal{L}_{\text{WM}}(\theta) + \mathcal{L}_{\text{EA}}(\theta) = \mathcal{L}_{\text{WM}}(\theta) + \beta_o \mathcal{L}_o(\theta) + \beta_e \mathcal{L}_e(\theta), \tag{9}$$

where $\mathcal{L}_{\text{WM}}(\theta)$ denotes the loss functions of the base world model except the loss for observations.

Table 1: Game scores and human normalized aggregate metrics on the 26 games of the Atari 100K benchmark. We highlight the highest and the second highest scores among all baselines in bold and with underscores, respectively. The results on Atari 100k are based on the established re-implementation of the original version of DreamerV3 (Hafner et al., 2023) in PyTorch using the default hyperparameters. We follow the official implementation of Simulus (Cohen et al., 2025) and reproduce the results based on the suggested hyperparameters. All results of our experiments are reported over 5 seeds.

| Game | Random | Human | REM | DIAMOND | DreamerV3 | HarmonyDream | EADream(Ours) | Simulus | EASimulus(Ours) |
|---|---|---|---|---|---|---|---|---|---|
| Alien | 227.8 | 7127.7 | 607.2 | 744.1 | 1024.9 | **1179.3** | 776.4 | 691.8 | 740.0 |
| Amidar | 5.8 | 1719.5 | 95.3 | **225.8** | 130.8 | 166.3 | 144.2 | 104.7 | 148.2 |
| Assault | 222.4 | 742.0 | 1764.2 | 1526.4 | 723.6 | 701.7 | 883.4 | 1528.8 | **2112.7** |
| Asterix | 210.0 | 8503.3 | 1637.5 | **3698.5** | 1024.2 | 1260.2 | 1096.9 | 1477.3 | 1590.3 |
| Bank Heist | 14.2 | 753.1 | 19.2 | 19.7 | **1018.9** | 627.1 | 742.6 | 283.1 | 524.9 |
| Battle Zone | 2360.0 | 37187.5 | 11826 | 4702.0 | 11246.7 | 11563.3 | 13372.0 | 10142.3 | **14348.1** |
| Boxing | 0.1 | 12.1 | 87.5 | 86.9 | 84.8 | 86.0 | 85.4 | **91.2** | 87.4 |
| Breakout | 1.7 | 30.5 | 90.7 | 132.5 | 26.9 | 34.9 | 71.8 | 153.4 | **236.5** |
| Chopper Command | 811.0 | 7387.8 | 2561.2 | 1369.8 | 709.7 | 627.0 | 904.0 | **4593.6** | 3446.3 |
| Crazy Climber | 10780.5 | 35829.4 | 76547.6 | **99167.8** | 81414.7 | 54687.3 | 89038.6 | 77456.3 | 79274.4 |
| Demon Attack | 152.1 | 1971.0 | **5738.6** | 288.1 | 226.5 | 267.0 | 152.2 | 5045.9 | 5081.9 |
| Freeway | 0.0 | 29.6 | 32.3 | **33.3** | 9.5 | 0.0 | 0.0 | 31.2 | 32.4 |
| Frostbite | 65.2 | 4334.7 | 240.5 | 274.1 | 251.7 | **1937.9** | 692.6 | 264.2 | 579.5 |
| Gopher | 257.6 | 2412.5 | 5452.4 | 5897.9 | 4074.9 | **9564.7** | 4415.8 | 4414.2 | 5300.0 |
| Hero | 1027.0 | 30826.4 | 6484.8 | 5621.8 | 4650.9 | **9865.3** | 8801.8 | 6626.9 | 7273.4 |
| James Bond | 29.0 | 302.8 | 391.2 | 427.4 | 331.8 | 327.8 | 337.2 | **677.3** | 612.4 |
| Kangaroo | 52.0 | 3035.0 | 467.6 | 5382.2 | 3851.7 | 5237.3 | 3875.6 | **7858.0** | 7569.6 |
| Krull | 1598.0 | 2665.5 | 4017.7 | 8610.1 | 7796.6 | 7784.0 | 8729.6 | 7385.7 | **9065.0** |
| Kung Fu Master | 258.5 | 22736.3 | **25172.2** | 18713.6 | 18917.1 | 22131.7 | 23434.6 | 20463.1 | 19456.2 |
| Ms Pacman | 307.3 | 6951.6 | 962.5 | 1958.2 | 1813.3 | **2663.3** | 1580.7 | 1331.9 | 1430.5 |
| Pong | -20.7 | 14.6 | 18.0 | **20.4** | 17.1 | 20.0 | 20.1 | 19.1 | 20.0 |
| Private Eye | 24.9 | 69571.3 | 99.6 | **114.3** | 47.4 | -198.6 | -472.5 | 58.7 | 100.0 |
| Qbert | 163.9 | 13455.0 | 743 | **4499.3** | 873.2 | 1863.3 | 1664.4 | 2234.0 | 3649.2 |
| Road Runner | 11.5 | 7845.0 | 14060.2 | 20673.2 | 14478.3 | 12478.3 | 12518.6 | **26325.0** | 20748.5 |
| Seaquest | 68.4 | 42054.7 | 1036.7 | 551.2 | 479.1 | 540.7 | 557.9 | **1602.7** | 1405.9 |
| Up N Down | 533.4 | 11693.2 | 3757.6 | 3856.3 | 20183.2 | 10007.1 | **28408.2** | 3609.9 | 5742.4 |
| #Superhuman(↑) | 0 | N/A | **12** | 11 | 10 | 9 | **12** | **12** | **12** |
| Mean(↑) | 0.000 | 1.000 | 1.222 | 1.459 | 1.150 | 1.200 | 1.290 | 1.609 | **1.818** |
| Median(↑) | 0.000 | 1.000 | 0.280 | 0.373 | 0.575 | 0.634 | 0.651 | 0.737 | **0.773** |
| IQM(↑) | 0.000 | 1.000 | 0.673 | 0.641 | 0.521 | 0.561 | 0.593 | 0.913 | **1.004** |
| Optimality Gap(↓) | 1.000 | 0.000 | 0.482 | 0.480 | 0.501 | 0.473 | 0.474 | 0.424 | **0.394** |

Table 2: Scores achieved across ten challenging tasks from DeepMind Control Suite with a budget of 500K interactions. We highlight the highest and the second highest scores among all baselines in bold and with underscores, respectively.

| Task | CURL | DrQ-v2 | TD-MPC2 | DreamerV3 | EADream(Ours) |
|---|---|---|---|---|---|
| Acrobot Swingup | 5.1 | 128.4 | 241.3 | 210.0 | **452.1** |
| Cartpole Swingup Sparse | 236.2 | 706.9 | 790.0 | 792.9 | **825.8** |
| Cheetah Run | 474.3 | 691.0 | 537.3 | 728.7 | **874.3** |
| Finger Turn Easy | 338.0 | 448.4 | 820.8 | 787.7 | **916.6** |
| Finger Turn Hard | 215.6 | 220.0 | 865.6 | 810.8 | **921.5** |
| Hopper Hop | 152.5 | 189.9 | 267.6 | **369.6** | 311.5 |
| Hopper Stand | 786.8 | 893.0 | 790.3 | 900.6 | **926.2** |
| Quadruped Run | 141.5 | 407.0 | 283.1 | 352.3 | **648.7** |
| Quadruped Walk | 123.7 | **660.3** | 323.5 | 352.6 | 580.3 |
| Walker Run | 376.2 | 517.1 | 671.9 | 757.8 | **784.8** |
| Mean(↑) | 285.0 | 486.2 | 559.2 | 606.3 | **723.8** |
| Median(↑) | 225.9 | 482.8 | 604.6 | 743.3 | **805.3** |

## 4 EXPERIMENTS

We aim to evaluate the effectiveness and the generality of our proposed framework in this section. Details for benchmarks and baselines are included in Section 4.1. To demonstrate the adaptability of our event-aware representation learning, we conduct experiments on two outstanding world models that were based on quite different architectures, *i.e.*, DreamerV3 and Simulus, yielding EADream and EASimulus, respectively. A comprehensive evaluation of these world models is presented in Section 4.2. Ablation studies of the key elements proposed for EAWM are shown in Section 4.3.

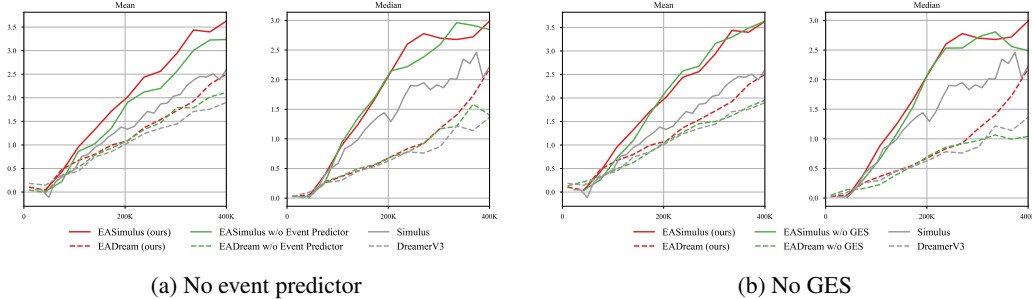

(a) No event predictor        (b) No GES

Figure 4. Ablation studies on key components of EAWMs with 5 random seeds over 6 Atari games: *Assault*, *Breakout*, *Gopher*, *Krull*, and *Ms Pacman*, *Up N Down*. The results show Simulus with the solid lines and EADream with the dashed lines.

## 4.1 BENCHMARKS AND BASELINES

**Atari 100K Benchmark** is comprised of 26 different Atari video games (Bellemare et al., 2013) across a diverse range of genres. The benchmark challenges general algorithms to sample-efficient learning within 100K interactions with 4 repeated actions, equivalent to 2 hours of human gameplay. The scores for each task are computed by conducting 100 evaluation episodes at the end of training. Standard measurement for a game is human-normalized score (HNS) (Mnih et al., 2015), calculated as $\text{HNS} = \frac{\text{agent score} - \text{random score}}{\text{human score} - \text{random score}}$.

**Craftax** is a recently released benchmark (Matthews et al., 2024) that features an open-world environment like Minecraft. It challenges agents' capacity to conduct deep exploration, craft resources, and survive in complex scenarios. Multimodal observations consist of an egocentric map and state information.

**DeepMind Control Suite** provides a collection of classical continuous control tasks (Tunyasuvunakool et al., 2020) widely used in robotics and reinforcement learning research. In our experiments, we restrict algorithm inputs to high-dimensional image observations. Following established convention (Hansen et al., 2024), the number of environment steps is 1M, which amounts to 500K interactions with 2 repeated actions. We evaluate EADream on the hard task group introduced by Hubert et al. (2021), which are listed in Table 2.

**DMC-GB2**, as an extension of the DMControl Generalization Benchmark (Hansen & Wang, 2021), consists of six continuous control tasks (Almuzairee et al., 2024). Agents are trained in a fixed environment but evaluated across visually distinct test environments. We evaluate our EADream on the hard test set, including randomized colors and random substitutions of the original background for videos. Please refer to Appendix I for visualizations of various test environments.

We choose competent baselines for the three benchmarks. On Atari 100K, besides DreamerV3 (Hafner et al., 2025) and Simulus (Cohen et al., 2025), world models include REM (Cohen et al., 2024), DIAMOND (Alonso et al., 2024), and HarmonyDream (Ma et al., 2024). On Craftax 1M, we include the best model of TWM (Best-TWM) (Dedieu et al., 2025). On DeepMind Control 500K, apart from DreamerV3 and TD-MPC2 (Hansen et al., 2024), recent model-free RL methods, CURL (Laskin et al., 2020) and DrQ-v2 (Yarats et al., 2022), are also included. On DMC-GB2, we include SVEA (Hansen et al., 2021) and SADA (Almuzairee et al., 2024), specially designed algorithms on the benchmark that need pairs of original images and augmented images. For a fair comparison (Zhang et al., 2024b; Cohen et al., 2025), here we exclude lookahead search methods. Nevertheless, lookahead search techniques can be integrated with EAWMs at the expense of computational burden.

## 4.2 IMPLEMENTATION DETAILS AND RESULTS

**EADream** Our observation predictor $\hat{\mathbf{o}}_t \sim p_\theta(\hat{\mathbf{o}}_t | \mathbf{h}_t, \hat{\mathbf{z}}_t)$ is slightly different from the observation decoder $\hat{\mathbf{o}}_t \sim p_\theta(\hat{\mathbf{o}}_t | \mathbf{h}_t, \mathbf{z}_t)$ in DreamerV3 (Hafner et al., 2025). EADream predicts future observations directly from the prior states $\mathbf{z}_t$, and thus we name this dynamic model RSSM-OP. To achieve computututational efficiency, we implement a simple form of GES: $g(\alpha_t^{(m)}, \alpha_{\text{thr}}^{(m)}) = \mathbb{I}(\alpha_t^{(m)} < \alpha_{\text{thr}}^{(m)})$.

Please refer to Appendix A for more details. As Table 2 displays, EADream reaches a mean score of 723.8 and a median score of 805.3, setting new state-of-the-art results among all RL methods on these challenging tasks from DeepMind Control Suite. On DMC-GB2, EADream outperforms DreamerV3 by a large margin, highlighting its robustness to visual distractors and generalization across visual variations. Beyond our expectations, EADream even outperforms SADA, an algorithm specifically designed for the benchmark. Unlike SADA, EAWM does not rely on paired original and augmented images, demonstrating that strong generalization can be achieved without additional supervision.

**EASimulus** Integrating Simulus into EAWM is straightforward, requiring only the integration of the event predictor and GES. Specifically, the output of the GES increases as events become sparser, allowing them to be better highlighted: $g(\alpha_t^{(m)}, \alpha_{\text{thr}}^{(m)}) = \mathbb{I}(\alpha_t^{(m)} < \alpha_{\text{thr}}^{(m)})/\text{arsinh}\left[\text{clip}\left(\frac{\alpha_t^{(m)}}{\alpha_{\text{thr}}^{(m)}}, \epsilon_\alpha, 1\right)\right]$,

where $\epsilon_\alpha > 0$ is a coefficient to smooth the curve of $\alpha_t^{(m)}$ and stablize the training process. Other details are provided in Appendix B. Remarkably, EASimulus obtains a mean human-normalized score of 1.818, setting a new record and reaching a superhuman IQM score for the first time among MBRL methods. In addition, EAWM provides a boost to the performance of the state-of-the-art method Simulus and reaches a score of 7.23% in Craftax 1M, indicating that multi-modal event awareness facilitates policy learning.

## 4.3 Ablation Studies

We discuss the effectiveness of key components of EAWM. We randomly select 6 games from Atari, as illustrated in Figure 4, and 4 tasks from DeepMind Control Suite, as shown in Figure 5.

**No Event Predictor** In this case, the GES influences the loss function for observation prediction in Equation 8. Figure 4a shows the decrease in mean HNS scores of both world models by about 0.4 without the event predictor. We observe that representation learning from event prediction plays an essential role in environments where events provide sufficient information for reward prediction, such as *Breakout* and *Krull*. Furthermore, these results show how kinetic features reshape and accelerate policy learning.

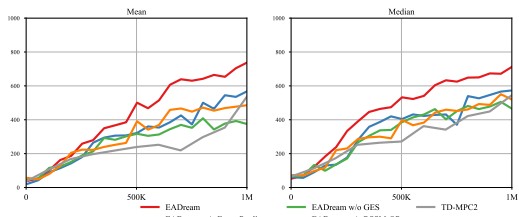

Figure 5. Ablation studies on key components of EAWM on DeepMind Control Suite.

**No GES** The GES stabilizes the training process of world models and provides robust event-aware representations for policy learning. As shown in Figure 4b, the median HNS scores grow slowly and display large fluctuations in the absence of GES. This stability is particularly critical in continuous control robotics tasks, where recovering from a single erroneous action often requires executing a long sequence of corrective steps.

**Without Observation Prediction** We notice a performance drop if EAWM is configured to make observation reconstruction. Specifically, the mean score of four tasks declines from 737.2 to 519.5, as shown by the orange line in Figure 5. In addition, we conduct an experiment on DreamerV3 with RSSM-OP in Table 13, demonstrating that the performance improvement of EAWM mainly stems from the joint modeling of observations and events, not from RSSM-OP alone. These results imply that event prediction and observation prediction are tightly coupled in event-aware representation learning.

## 5 Related Work

**Model-based Reinforcement Learning** Sample efficiency has become increasingly important in reinforcement learning, especially for real-world applications (Sutton, 1991; Hafner, 2022). The first world model (Ha & Schmidhuber, 2018) showed that combining latent dynamics with a generative observation model enables agents to plan through imagination, sparking broad interest in MBRL (Moerland et al., 2023). Dreamer, a notable series of methods (Hafner et al., 2019a; 2020; 2025), which is based on the recurrent state-space model, constructed dynamics models in latent space. Descendants

of RSSM (Gumbsch et al., 2023; Sun et al., 2024), were proposed to learn hierarchical and robust latent representations. Encouraged by the huge success of Transformer architecture (Vaswani, 2017), several works (Chen et al., 2022; Micheli et al., 2023; Seo et al., 2023; Robine et al., 2023; Zhang et al., 2024b; Burchi & Timofte, 2025) attempted to use a transformer-based world model to learn the dynamics in environments. In parallel, TD-MPC2 (Hansen et al., 2024) predicts future states rather than observations, achieving strong performance in state space but struggling when applied directly to image-based environments. Following REM Cohen et al. (2024), Simulus Cohen et al. (2025) surpasses previous methods on Atari 100K and Craftax 1M benchmarks. DyMoDreamer (Zhang et al., 2025) is the most closely related work to our approach, which enhances DreamerV3 (Hafner et al., 2025) by encoding differential observations obtained via an inter-frame differencing mask, demonstrating the effectiveness of dynamic features. Instead of treating inter-frame differences as inputs, we train EAWM to predict both future observations and future events. Furthermore, our method is totally distinct from the above methods — EAWM is not a standalone architecture, but a general framework that can endow different world models with a general understanding of more abstract events.

**Representation Learning for Reinforcement Learning** Representation learning is key to sample-efficient deep RL (Nikishin et al., 2022; Wang et al., 2024a), yet non-stationary targets and bootstrapping destabilize training (Voelcker et al., 2024). Early methods such as UNREAL (Jaderberg et al., 2017) introduced specialized auxiliary tasks, while contrastive methods (Laskin et al., 2020; Stooke et al., 2021; You et al., 2022; Eysenbach et al., 2022) and image augmentations (Yarats et al., 2021; 2022) improved consistency and robustness. Recent work (Wu et al., 2023; Kim et al., 2024) also explores pretraining models with abundant videos for downstream visual control tasks. With increasing empirical (Ni et al., 2024; Fang & Stachenfeld, 2024; Kwon et al., 2025) and theoretical (Uehara et al., 2022; Liu et al., 2023) evidence proving the benefit of using predictive auxiliary tasks, next-observation prediction has become a common choice for modern MBRL. However, these methods often struggle to extract the task-relevant structure in high-dimensional inputs (Zhang et al., 2024a). Alternatives like depth (Poudel et al., 2024) or motion prediction help, but they remain limited to static visual inputs and require additional labels. In contrast, our work introduces event prediction as a label-free auxiliary task, enabling agents to capture task-relevant dynamics and structure without reliance on manual annotations.

## 6 CONCLUSION

In this paper, we have introduced EAWM, a general framework that enables world models to capture key events from the environments. EAWMs master tasks across different domains for control, be it discrete or continuous. Specifically, EAWM achieves a 13%, 10%, 19%, and 45% performance boost on Atari 100K, Craftax 1M, DeepMind Control 500K, and DMC-GB2 500K, respectively. Moreover, EAWMs have established new state-of-the-art results across these benchmarks. Overall, EAWM paves the way for building RL agents that react to events rather than static images.

We identify three limitations of our work for future research. First, as this is the initial attempt to integrate event segmentation into world models for policy learning, our implementation of GES is deliberately simple for efficiency. Prior approaches to event boundary detection, such as using changes in reconstruction error of video snippets (Wang et al., 2024b), are more specialized but contain substantial redundancy (Zheng et al., 2024). Future work may consider modeling the function of GES directly with neural networks to achieve both efficiency and expressiveness. Second, although EADream and EASimulus are trained with fixed hyperparameters across domains, developing a unified model capable of solving multiple tasks by effectively sharing common knowledge is still challenging (Sridhar et al., 2025). Finally, another promising direction is to combine EAWM with large pretrained vision-language models, which could enhance generalization and enable more flexible grounding across modalities (Contributors, 2025; Chen et al., 2025; Yang et al., 2025).

## ACKNOWLEDGMENT

This work was partially supported by the National Natural Science Foundation of China (No. 62293545, 62425117, 62301299, and 62506205), the China Postdoctoral Science Foundation (No. 2025T180426), the Postdoctoral Fellowship Program of CPSF (No. GZB20250393).

REPRODUCIBILITY STATEMENT

We have made every effort to ensure the reproducibility of our work. We include theoretical proofs of our EADream in Appendix A.2 and A.3. The architectures of EADream and EASimulus are fully specified in Appendix A.4 and B.2, with hyperparameters provided in Appendix A.5 and B.3. Pseudocode for the training protocol of EAWM is included in Appendix C, and the configurations of multi-modal event prediction are in Appendix D. Extended results, including Atari 1M and all tasks from DeemMind Control Suite, are provided in Appendix J and H. More experimental settings, including baselines and computational resources, are described in Section E and M. For transparency, training curves are shown in the Appendix K and L. In addition, results and visualizations are included in the supplementary materials. We will release our code and checkpoints to facilitate replication of our experiments.

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

APPENDICES

# A EADREAM

## A.1 FRAMEWORK

**Components** With the length of trajectory segments set to 1, segments of observations $\mathbf{O}_t$ and actions $\mathbf{A}_t$ in Section 3.1 degenerate into $\mathbf{o}_t$ and $\mathbf{a}_t$, respectively. DreamerV3 Hafner et al. (2025) uses the hidden state $\mathbf{h}_t$ as the output $\mathbf{y}_t$ of the sequence model in Equation equation 3. That is to say, $\mathbf{y}_t = \mathbf{h}_t$. Consequently, the formulation of each component turns into:

$$
\begin{aligned}
\text{DreamerV3} \begin{cases}
\text{Sequence Model:} & \mathbf{h}_t = \mathbf{F}_\theta(\mathbf{h}_{t-1}, \mathbf{z}_{t-1}, \mathbf{a}_{t-1}) \\
\text{Encoder Network:} & \mathbf{z}_t \sim q_\theta(\mathbf{z}_t \,|\, \mathbf{o}_t, \mathbf{h}_t) \\
\text{Dynamics Predictor:} & \hat{\mathbf{z}}_t \sim p_\theta(\hat{\mathbf{z}}_t \,|\, \mathbf{h}_t) \\
\text{Reward Predictor:} & \hat{r}_t \sim p_\theta(\hat{r}_t | \mathbf{h}_t, \mathbf{z}_t) \\
\text{Ending Predictor:} & \hat{e}_t \sim p_\theta(\hat{e}_t | \mathbf{h}_t, \mathbf{z}_t) \\
\text{Observation Predictor:} & \hat{\mathbf{o}}_t \sim p_\theta(\hat{\mathbf{o}}_t | \mathbf{h}_t, \hat{\mathbf{z}}_t) \\
\end{cases} \\
\text{EA} \begin{cases}
\text{Event Predictor:} & \hat{\mathbf{e}}_t \sim p_\theta(\hat{\mathbf{e}}_t | \mathbf{h}_t, \hat{\mathbf{z}}_t, \mathbf{z}_t) \\
\text{Generic Event Segmentor:} & \hat{\mathbf{g}}_t = g_\theta(\mathbf{o}_t, \hat{\mathbf{o}}_t, \mathbf{e}_t) \\
\end{cases}
\end{aligned}
\tag{10}
$$

The first five components above are the same as that of DreamerV3 Hafner et al. (2025). Notice that our observation predictor $\hat{\mathbf{o}}_t \sim p_\theta(\hat{\mathbf{o}}_t | \mathbf{h}_t, \hat{\mathbf{z}}_t)$ is different from the observation decoder of DreamerV3 (Hafner et al., 2025) denoted by $\hat{\mathbf{o}}_t \sim p_\theta(\hat{\mathbf{o}}_t | \mathbf{h}_t, \mathbf{z}_t)$. EADream predicts the next observations $\hat{\mathbf{o}}_t$ without access to them, while DreamerV3 makes a reconstruction of the observations from $\mathbf{z}_t$. Due to the subtle change in the formulation, it is necessary to reinterpret the related models and the training objectives proposed by DreamerV3, as we do in Appendix A.2 and Appendix A.3.

**Loss Functions** For DreamerV3, we implement a simple form of GES:

$$
g(\alpha_t^{(m)}, \alpha_{\text{thr}}^{(m)}) = \mathbb{I}(\alpha_t^{(m)} < \alpha_{\text{thr}}^{(m)}),
\tag{11}
$$

From Equation equation 8, the event-aware prediction loss for observations is:

$$
\mathcal{L}_t^{\text{o}}(\theta) = \boldsymbol{\epsilon}_t + \omega \mathbb{I}(\alpha_t^{(m)} < \alpha_{\text{thr}}^{(m)})(\mathbf{e}_t - \mathbf{1}) \odot \boldsymbol{\epsilon}_t,
\tag{12}
$$

where $\boldsymbol{\epsilon}_t = (\epsilon_{t,1}, ..., \epsilon_{t,N})$ and $\epsilon_{t,i}$ is the loss function of $o_{t,i}$ and $\hat{o}_{t,i}$. If every observation changes or the GES is disabled, then observation prediction loss $\mathcal{L}_t^{\text{o}}(\theta)$ is equivalent to the original loss for observations in DreamerV3.

## A.2 REINTERPRETATIONS OF THE REPRESENTATION MODEL AND DYNAMICS MODEL

Since the recurrent state space model with observation prediction can be seen as a Markov process as shown in Figure 6, the encoder can be formulated as $q(\mathbf{s}_{0:T-1}|\mathbf{a}_{0:T-1}, \mathbf{o}_{1:T}, \mathbf{o}_0) = \prod_{t=0}^{T-1} q(\mathbf{s}_t|\mathbf{o}_{\leq t}, \mathbf{a}_{<t}) = \prod_{t=0}^{T-1} q(\mathbf{s}_t|\mathbf{o}_t, \mathbf{h}_t)$, where $\mathbf{h}_t = \mathbf{h}(\mathbf{s}_{t-1}, \mathbf{a}_{t-1})$ and $\mathbf{s}_t = \mathbf{s}(\mathbf{h}_t, \mathbf{z}_t)$ are deterministic functions. Therefore, the distribution of $\mathbf{s}_t$ can be obtained if we know the distribution of the stochastic state $\mathbf{z}_t$. We parameterize the distribution of $\mathbf{z}_t$ via representation model $q_\theta(\mathbf{z}_t|\mathbf{o}_t, \mathbf{h}_t)$, where $\mathbf{h}_t = \mathbf{h}(\mathbf{s}_{t-1}, \mathbf{a}_{t-1})$ can be implemented as a recurrent neural network. Similarly, we can obtain $p(\mathbf{s}_t|\mathbf{s}_{t-1}, \mathbf{a}_{t-1})$ from $p(\mathbf{z}_t|\mathbf{s}_{t-1}, \mathbf{a}_{t-1}) = p(\mathbf{z}_t|\mathbf{h}_t)$, which necessitates the dynamics model $p_\theta(\hat{\mathbf{z}}_t|\mathbf{h}_t)$. Furthermore, we have $D_{\text{KL}}(q(\mathbf{s}_t|\mathbf{o}_t, \mathbf{h}_t)||p(\mathbf{s}_t|\mathbf{h}_t)) = D_{\text{KL}}(q(\mathbf{z}_t|\mathbf{o}_t, \mathbf{h}_t)||p(\mathbf{z}_t|\mathbf{h}_t))$ due to the implementation of $\mathbf{s}_t$ in DreamerV3, which is the concatenation of $\mathbf{h}_t$ and $\mathbf{z}_t$.

## A.3 PROOF OF TRAINING OBJECTIVES

Though (Hafner et al., 2019b) has proved an objective for world model training in Dreamer algorithms, it is unclear what the training objective is to learn a world model utilizing RSSM for observation prediction (RSSM-OP). A direct objective is to maximize the log-likelihood of video data. Therefore, we derive the Evidence Lower BOund (ELBO) of the log-likelihood conditioned on the action sequence and the first frame.

Since we want to predict the next frame conditioned on the current state and action, the latent dynamics model is $p(\mathbf{o}_{1:T}, \mathbf{s}_{0:T-1}|\mathbf{a}_{0:T-1}, \mathbf{o}_0) = \prod_{t=1}^{T} p(\mathbf{o}_t|\mathbf{s}_{t-1}, \mathbf{a}_{t-1}, \mathbf{o}_0) p(\mathbf{s}_{t-1}|\mathbf{s}_{t-2}, \mathbf{a}_{t-2}, \mathbf{o}_0) =$

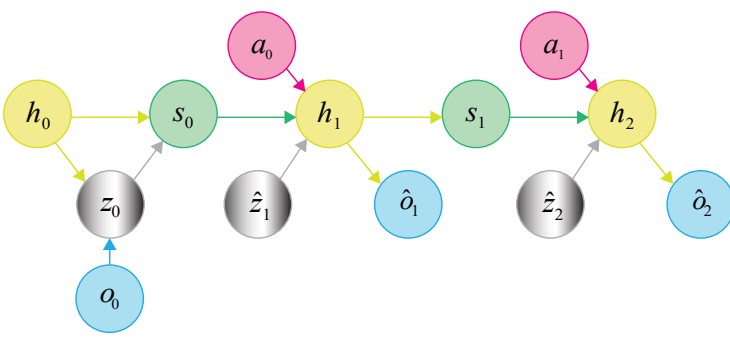

Figure 6. RSSM with observation prediction.

$\prod_{t=1}^{T} p(\mathbf{o}_t|\mathbf{s}_{t-1}, \mathbf{a}_{t-1}) p(\mathbf{s}_{t-1}|\mathbf{s}_{t-2}, \mathbf{a}_{t-2})$, where $p(\mathbf{s}_0|\mathbf{s}_{-1}, \mathbf{a}_{-1})$ is defined as $p(\mathbf{s}_0|\mathbf{o}_0)$. Accordingly, the variational posterior is $q(\mathbf{s}_{0:T-1}|\mathbf{a}_{0:T-1}, \mathbf{o}_{1:T}, \mathbf{o}_0) = \prod_{t=0}^{T-1} q(\mathbf{s}_t|\mathbf{o}_{\leq t}, \mathbf{a}_{<t})$, where we define $q(\mathbf{s}_0|\mathbf{o}_{\leq 0}, \mathbf{a}_{<0})$ as $q(\mathbf{s}_0|\mathbf{o}_0)$. Using importance weighting and Jensen's inequality, the ELBO of the likelihood of the image conditioned on the first frame and history of actions is:

$$\ln p(\mathbf{o}_{1:T}|\mathbf{a}_{0:T-1}, \mathbf{o}_0) \triangleq \ln \mathbb{E}_{p(\mathbf{s}_{0:T-1}|\mathbf{a}_{0:T-1}, \mathbf{o}_0)} \left[ \prod_{t=1}^{T} p(\mathbf{o}_t|\mathbf{s}_{t-1}, \mathbf{a}_{t-1}) \right]$$

$$= \ln \mathbb{E}_{q(\mathbf{s}_{0:T-1}|\mathbf{a}_{0:T-1}, \mathbf{o}_0)} \left[ \frac{\prod_{t=1}^{T} p(\mathbf{o}_t|\mathbf{s}_{t-1}, \mathbf{a}_{t-1}) p(\mathbf{s}_{t-1}|\mathbf{s}_{t-2}, \mathbf{a}_{t-2})}{q(\mathbf{s}_{0:T-1}|\mathbf{a}_{0:T-1}, \mathbf{o}_0)} \right]$$

$$= \ln \mathbb{E}_{q(\mathbf{s}_{0:T-1}|\mathbf{a}_{0:T-1}, \mathbf{o}_0)} \left[ \prod_{t=1}^{T} p(\mathbf{o}_t|\mathbf{s}_{t-1}, \mathbf{a}_{t-1}) p(\mathbf{s}_{t-1}|\mathbf{s}_{t-2}, \mathbf{a}_{t-2})/q(\mathbf{s}_{t-1}|\mathbf{o}_{<t}, \mathbf{a}_{<t-1}) \right]$$

$$\geq \mathbb{E}_{q(\mathbf{s}_{0:T-1}|\mathbf{a}_{0:T-1}, \mathbf{o}_0)} \left[ \sum_{t=1}^{T} \ln p(\mathbf{o}_t|\mathbf{s}_{t-1}, \mathbf{a}_{t-1}) + \ln p(\mathbf{s}_{t-1}|\mathbf{s}_{t-2}, \mathbf{a}_{t-2}) - \ln q(\mathbf{s}_{t-1}|\mathbf{o}_{<t}, \mathbf{a}_{<t-1}) \right]$$

$$= \sum_{t=1}^{T} \mathbb{E}_{q(\mathbf{s}_{t-1}|\mathbf{o}_{<t}, \mathbf{a}_{<t-1})} \left[ \ln p(\mathbf{o}_t|\mathbf{s}_{t-1}, \mathbf{a}_{t-1}) \right]$$

$$- \sum_{t=1}^{T} \mathbb{E}_{q(\mathbf{s}_{t-2}|\mathbf{o}_{<t-1}, \mathbf{a}_{<t-2})} \left[ D_{\text{KL}} \left( q(\mathbf{s}_{t-1}|\mathbf{o}_{<t}, \mathbf{a}_{<t-1}) || p(\mathbf{s}_{t-1}|\mathbf{s}_{t-2}, \mathbf{a}_{t-2}) \right) \right].$$

For $T \rightarrow \infty$, we always minimize the KL divergence of the latent dynamics models $p(\mathbf{s}_{t-1}|\mathbf{s}_{t-2}, \mathbf{a}_{t-2})$ and the variational posterior $q(\mathbf{s}_{t-1}|\mathbf{o}_{<t}, \mathbf{a}_{<t-1})$. Set $t' = t - 1$ and then substitute $t'$ for $t$. The second term will be

$$\sum_{t=0}^{\infty} \mathbb{E}_{q(\mathbf{s}_{t-1}|\mathbf{o}_{<t}, \mathbf{a}_{<t-1})} \left[ D_{\text{KL}} \left( q(\mathbf{s}_t|\mathbf{o}_{\leq t}, \mathbf{a}_{<t}) || p(\mathbf{s}_t|\mathbf{s}_{t-1}, \mathbf{a}_{t-1}) \right) \right].$$

We sample a batch from episodes, and minimizing the KL divergence helps improve the prediction of future frames across different batches. Therefore, the modified objective is to maximize

$$\sum_{t=1}^{T} \left( \mathbb{E}_{q(\mathbf{s}_{t-1}|\mathbf{o}_{<t}, \mathbf{a}_{<t-1})} \left[ \ln p(\mathbf{o}_t|\mathbf{s}_{t-1}, \mathbf{a}_{t-1}) - D_{\text{KL}} \left( q(\mathbf{s}_t|\mathbf{o}_{\leq t}, \mathbf{a}_{<t}) || p(\mathbf{s}_t|\mathbf{s}_{t-1}, \mathbf{a}_{t-1}) \right) \right] \right)$$
$$- D_{\text{KL}}(q(\mathbf{s}_0|\mathbf{o}_0) || p(\mathbf{s}_0|\mathbf{o}_0)).$$

In fact, we can set $q(\mathbf{s}_0|\mathbf{o}_0) \equiv p(\mathbf{s}_0|\mathbf{o}_0)$ to save the hassle of dealing with inputs with different dimensions. Therefore, our training objective for observation prediction is:

$$\sum_{t=1}^{T} \left( \mathbb{E}_{q(\mathbf{s}_{t-1}|\mathbf{o}_{<t}, \mathbf{a}_{<t-1})} \left[ \ln p(\mathbf{o}_t|\mathbf{s}_{t-1}, \mathbf{a}_{t-1}) - D_{\text{KL}} \left( q(\mathbf{s}_t|\mathbf{o}_{\leq t}, \mathbf{a}_{<t}) || p(\mathbf{s}_t|\mathbf{s}_{t-1}, \mathbf{a}_{t-1}) \right) \right] \right), \quad (13)$$

which takes the same form as the training objective for observation reconstruction in DreamerV3 (Hafner et al., 2025), which is listed below for comparison:

$$\sum_{t=1}^{T} \left( \mathbb{E}_{q(\mathbf{s}_{t-1}|\mathbf{o}_{<t}, \mathbf{a}_{<t-1})} \left[ \ln p(\mathbf{o}_t|\mathbf{s}_t) - D_{\mathrm{KL}} \left( q(\mathbf{s}_t|\mathbf{o}_{\leq t}, \mathbf{a}_{<t}) || p(\mathbf{s}_t|\mathbf{s}_{t-1}, \mathbf{a}_{t-1}) \right) \right] \right), \qquad (14)$$

### A.4 ARCHITECTURE

The representation model in DreamerV3 (Hafner et al., 2025) consists of the observation encoder and the representation predictor. Table 3 shows the process of the image encoder to obtain embeddings of an image, since we focus on dealing with high-dimensional visual inputs for experiments on EADream. Here, $r$ is the reduction ratio of the ChannelAttention submodule and $k$ is the kernel size of the SpatialAttention submodule. The representation predictor takes as input the embeddings and the deterministic hidden states $\mathbf{h}_t$ to obtain the posterior stochastic hidden states $\mathbf{z}_t$, as described in Table 4. LayerNorm denotes layer normalization (Ba, 2016), and SiLU is short for sigmoid-weighted linear units, an activation function which is formulated as $\mathrm{SiLU}(x) = \frac{x}{1+e^{-x}}$.

Table 3: Structure of the observation encoder

| Stage name | Output size | Submodule |
|---|---|---|
| CBAM | $64 \times 64$ | ChannelAttention, $r = 1$
SpatialAttention, $k = 3$ |
| Conv1 | $32 \times 32$ | $4 \times 4$, 32, stride 2
LayerNorm + SiLU |
| Conv2 | $16 \times 16$ | $4 \times 4$, 64, stride 2
LayerNorm + SiLU |
| Conv3 | $8 \times 8$ | $4 \times 4$, 128, stride 2
LayerNorm + SiLU |
| Conv4 | $4 \times 4$ | $4 \times 4$, 256, stride 2
LayerNorm + SiLU |
| CBAM | $4 \times 4$ | ChannelAttention, $r = 2$
SpatialAttention, $k = 1$ |
| Flatten | 4096 | the embeddings of image |

Table 4: Structure of the representation predictor

| Stage name | Output size | Submodule |
|---|---|---|
| Inputs | $4096 + N_{\mathrm{deter}}$ | Concatenate $h_t$ and the embeddings of image |
| FC1 | $N_{\mathrm{hid}}$ | Linear
LayerNorm + SiLU |
| FC2 | $Z_{\mathrm{num}} \times Z_{\mathrm{class}}$ | Linear
LayerNorm + SiLU
Reshape |

Both the observation predictor and the event predictor are expected to output tensors of the same size. To that end, we implement similar networks for observation and event prediction, as depicted in Table 5 and Table 6. Table 7 displays MLP structures of the reward predictor and continuation predictor.

Table 5: Structure of the observation predictor

| Stage name | Output size | Submodule |
|---|---|---|
| Inputs | $N_{\text{stoch}} + N_{\text{deter}}$ | Concatenate $h_t$ and $\hat{z}_t$ |
| FC1 | $4 \times 4$ | Linear
Reshape into tensors of 256 channels |
| Deconv1 | $8 \times 8$ | $4 \times 4$, 128, stride 2
LayerNorm + SiLU |
| Deconv2 | $16 \times 16$ | $4 \times 4$, 64, stride 2
LayerNorm + SiLU |
| Deconv3 | $32 \times 32$ | $4 \times 4$, 32, stride 2
LayerNorm + SiLU |
| Deconv4 | $64 \times 64$ | $4 \times 4$, 3, stride 2 |

Table 6: Structure of the event predictor

| Stage name | Output size | Submodule |
|---|---|---|
| Inputs | $2N_{\text{stoch}} + N_{\text{deter}}$ | Concatenate $h_t$ and $\hat{s}_t$ |
| FC1 | $4 \times 4$ | Linear
Reshape into tensors of 256 channels |
| Deconv1 | $8 \times 8$ | $4 \times 4$, 128, stride 2
LayerNorm + SiLU |
| Deconv2 | $16 \times 16$ | $4 \times 4$, 64, stride 2
LayerNorm + SiLU |
| Deconv3 | $32 \times 32$ | $4 \times 4$, 32, stride 2
LayerNorm + SiLU |
| Deconv4 | $64 \times 64$ | $4 \times 4$, 1, stride 2
Sigmoid |

Table 7: Reward predictor and continuation predictor

| Details of MLP | Reward predictor | continuation predictor |
|---|---|---|
| Inputs | $N_{\text{stoch}} + N_{\text{deter}}$ | $N_{\text{stoch}} + N_{\text{deter}}$ |
| Hidden units | $N_{\text{unit}}$ | $N_{\text{unit}}$ |
| Outputs units | 255 | 1 |
| Activation function | SiLU | SiLU |
| Normalization | LayerNorm | LayerNorm |
| Layers | 2 | 2 |

## A.5 HYPERPARAMETERS

Table 8 shows the hyperparameters of EADream. These hyperparameters are fixed on both Atari 100K and DeepMind Control benchmarks.

Table 8: Hyperparameters in our world model, EADream. $N_{\text{stoch}}$ de facto denotes the dimension of the flattened version of $\mathbf{z}_t$. That is to say, $N_{\text{stoch}} = Z_{\text{num}} \cdot Z_{\text{class}}$ for discrete representations, which is our choice. $N_{\text{stoch}} = Z_{\text{num}}$ when continuous representations are applied.

| Type | Hyperparameter | Value |
|---|---|---|
| General | Image size | $64 \times 64 \times 3$ |
| | Batch size | 16 |
| | Batch length $T$ | 64 |
| | Gradient Clipping | 1000 |
| | Discount factor $\gamma$ | 0.997 |
| | Lambda $\lambda$ | 0.95 |
| World model | Number of stochastic variables $Z_{\text{num}}$ | 64 |
| | Classes per stochastic variable $Z_{\text{class}}$ | 32 |
| | Number of deterministic units $N_{\text{deter}}$ | 512 |
| | Number of stochastic units $N_{\text{stoch}}$ | 2048 |
| | Number of MLP units $N_{\text{unit}}$ | 512 |
| | Number of RSSM units $N_{\text{hid}}$ | 512 |
| | Imagination horizon | 15 |
| | First frame prediction | False |
| | Event prediction | True |
| | Updatation per interaction | 1 |
| | Harmonizers | True |
| | Optimizer | AdamW |
| | AdamW episilon $\epsilon$ | $1 \times 10^{-8}$ |
| | AdamW betas $(\beta_1, \beta_2)$ | $(0.9, 0.999)$ |
| | Learning rate | $1 \times 10^{-4}$ |
| | Gradient clipping | 1000 |
| Event-Aware Mechanism | Event prediction weight $\beta_{\text{e}}$ | 0.5 |
| | Observation prediction weight $\beta_{\text{o}}$ | 1.0 |
| | Focal loss alpha $\alpha$ | 0.15 |
| | Focal loss gamma $\gamma$ | 4 |
| | Event-aware weight $\omega$ | 0.5 |

## B EASIMULUS

### B.1 FRAMEWORK

**Components** Simulus (Cohen et al., 2025) is a modular world model that separates optimizations of the encoder-decoder network and the other parts of the world model. The sequence model of Simulus is a retentive network (Sun et al., 2023) augmented with parallel observation prediction (Cohen et al., 2024). We can also derive EASimulus from Equation equation 3 by omitting $\mathbf{y}_t$ in the representation model and making predictions depending only on the output of the sequence model $\mathbf{y}_t$:

$$
\begin{aligned}
\text{Simulus} &\begin{cases}
\text{Sequence Model:} & \mathbf{h}_t, \mathbf{y}_t = \mathbf{F}_\theta(\mathbf{h}_{t-1}, \mathbf{Z}_{t-1}, \mathbf{A}_{t-1}) \\
\text{Representation Model:} & \mathbf{z}_t \sim q_\theta(\mathbf{z}_t \,|\mathbf{o}_t) \\
\text{Dynamics Predictor:} & \hat{\mathbf{z}}_t \sim p_\theta(\hat{\mathbf{z}}_t \,|\mathbf{y}_t) \\
\text{Reward Predictor:} & \hat{r}_t \sim p_\theta(\hat{r}_t|\mathbf{y}_t) \\
\text{Continuation Predictor:} & \hat{c}_t \sim p_\theta(\hat{c}_t|\mathbf{y}_t) \\
\text{Observation Predictor:} & \hat{\mathbf{o}}_t \sim p_\theta(\hat{\mathbf{o}}_t|\hat{\mathbf{z}}_t)
\end{cases} \\
\text{EA} &\begin{cases}
\text{Event Predictor:} & \hat{\mathbf{e}}_t \sim p_\theta(\hat{\mathbf{e}}_t|\mathbf{y}_t, \mathbf{y}_{t+1}) \\
\text{Generic Event Segmentor:} & \hat{\mathbf{g}}_t = g_\theta(\mathbf{o}_t, \hat{\mathbf{o}}_t, \mathbf{e}_t)
\end{cases}
\end{aligned}
\tag{15}
$$

In Simulus, the representation model and the observation predictor form a pair of an encoder-decoder network. We follow their training protocol and jointly train the other parts of the world model besides the event predictor.

**Loss Functions** Intuitively, if events are sparsely distributed, agents should pay more attention to these events. In EASimulus, we implement GES as below:

$$
g(\alpha_t^{(m)}, \alpha_{\text{thr}}^{(m)}) = \frac{\mathbb{I}(\alpha_t^{(m)} < \alpha_{\text{thr}}^{(m)})}{\text{arsinh}\left[\text{clip}\left(\frac{\alpha_t^{(m)}}{\alpha_{\text{thr}}^{(m)}}, \epsilon_\alpha, 1\right)\right]},
\tag{16}
$$

where $\text{clip}(x, x_{\min}, x_{\max})$ is a function that limits $x$ within the range $[x_{\min}, x_{\max}]$, and $\text{arsinh}(x) = \ln(x + \sqrt{1 + x^2})$ is the inverse hyperbolic sine function. $\epsilon_\alpha > 0$ is a coefficient to smooth the curve of $\alpha_t^{(m)}$ and stablize the training process. Since the embeddings of current observations are learned independently in Simulus, we keep the same prediction loss of observations in EASimulus.

### B.2 ARCHITECTURE

Simulus (Cohen et al., 2025) is a token-based world model that maps observations $\mathbf{o}_t$ and actions $\mathbf{a}_t$ into tokens. We denote the number of tokens of observations as $K_o$, the number of tokens of observation as $K_a$, and the embedding dimension as $D$, where $K_o = \sum_{i=1}^{M} K_o^{(m)}$ and $K_o^{(m)}$ denotes the number of tokens of observations that are in the $m$-th modality. Then the number of tokens in one environment step is $K = K_o + K_a$. To our surprise, the simple structure in Table 9 is qualified to make an accurate prediction of events. In addition, Table 10 shows the structure of the reward predictor and the continuation predictor.

Table 9: Event segmentor in EASimulus for the $m$-th modality.

| Type of Data | Ordinal Data | Nominal Data |
|---|---|---|
| Inputs | $2 \times K_o^{(m)} \times D$ | $2 \times K_o^{(m)} \times D$ |
| Hidden units | $4D$ | $4D$ |
| Outputs units | 3 | 1 |
| Activation function | SiLU | SiLU |
| Normalization | LayerNorm | LayerNorm |
| Layers | 2 | 2 |

Table 10: Reward predictor and continuation predictor in EASimulus.

| Details of MLP | Reward Predictor | Continuation Predictor |
|---|---|---|
| Inputs | $K \times D$ | $K \times D$ |
| Hidden units | $2D$ | $2D$ |
| Outputs units | 129 | 1 |
| Activation function | SiLU | SiLU |
| Normalization | Symlog | LayerNorm |
| Layers | 2 | 2 |

## B.3 HYPERPARAMETERS

For EASimulus, we only tune hyperparameters that our methods introduce, as listed in Table 11:

Table 11: Hyperparameters in EASimulus.

| Type | Hyperparameter | Value |
|---|---|---|
| World model | First Observation prediction | False |
| | Event prediction | True |
| Event-Aware Mechanism | Event prediction weight $\beta_e$ | 1.0 |
| | Observation prediction weight $\beta_o$ | 1.0 |
| | Focal loss alpha $\alpha$ | 0.15 |
| | Focal loss gamma $\gamma$ | 4 |
| | Smoothing coefficient $\epsilon_\alpha$ | 0.0005 |

## C  ALGORITHM

The training process of EAWM is sketched out in Algorithm 1.

---
**Algorithm 1** EAWM Training
---
**Input:** An initialized replay buffer $\mathcal{D}$
**repeat**
    $\mathbf{o}_0, r_0, c_0 \leftarrow$ `env.reset()`
    Initialize parameters of `AutomatedProcess` using $\mathbf{o}_0$ (Section 3.3)
    **for** $t = 0$ **to** MAX_STEP **do**
        $\mathbf{a}_t \sim \pi(\mathbf{a}_t | \mathbf{o}_{\leq t}, \mathbf{a}_{<t})$
        $\mathbf{o}_{t+1}, r_{t+1}, c_{t+1} \leftarrow$ `env.step()`
        $\mathbf{e}_{t+1} \leftarrow$ `AutomatedProcess.generate($\mathbf{o}_{t+1}$)`
        **if** $c_{t+1} = 0$ **then**
            $t_m = t + 1$
            **break**
        **end if**
        Sample $B$ data of length $T$ from $\mathcal{D}$
        Predict $\{\hat{o}_t, \hat{m}_t, \hat{z}_t, z_t, \hat{r}_t, \hat{c}_t\}_{t=k}^{k+T-1}$ # Equation equation 3
        Compute loss from the base world model $\mathcal{L}_{\text{WM}}(\theta)$
        Compute event prediction loss with GES $\mathcal{L}_{\text{e}}(\theta)$ # Equation equation 7
        Compute event-aware prediction loss for observations $\mathcal{L}_{\text{o}}(\theta)$ # Equation equation 8
        Compute total loss $\mathcal{L}(\theta)$ # Equation equation 9
        Update parameters $\theta$
        Actor-critic learning in imagined trajectories
    **end for**
    $\mathcal{D}$.`add(` $\{\mathbf{o}_t, \mathbf{a}_t, \mathbf{e}_t, r_t, c_t\}_{t=0}^{t_m}$ `)`
**until** Training is stopped

---

## D  MULTIMODAL EVENTS

Table 12: Settings of related multimodal events

| Modality | Type | Symbol | Threshold $\alpha_{\text{thr}}^{(m)}$ | Weight Coefficient $\beta_{\text{e}}^{(m)}$ |
|---|---|---|---|---|
| Image | Visual Inputs | $e_{t,i}^{\text{vis}}$ | 0.5 | 1.0 |
| Continuous Variables | Ordinal Data | $e_{t,i}^{\text{ord}}$ | 1.0 | 0.1 |
| Categorical Variables | Nominal Data | $e_{t,i}^{\text{nom}}$ | 0.5 | 0.1 |
| 2D Categorical Variables | Nominal Data | $e_{t,i}^{\text{nom}}$ | 0.5 | 0.1 |

## E  BASELINES

### E.1  DREAMERV3

We used the default parameters and reproduced the results based on the implementation of DreamerV3 in PyTorch, which performs slightly better than the original official implementation of DreamerV3. In the original version of the paper (Hafner et al., 2023), DreamerV3 reported a HNS mean score of 112%.

### E.2  HARMONYDREAM

Since no hyperparameter is introduced in HarmonyDream (Ma et al., 2024), we implemented harmonizers following recommendations from the authors. We reproduced the results based on the

aforementioned implementation of DreamerV3 with harmonious loss, as suggested by the authors in their articles.

### E.3  TD-MPC2

Results for eight out of ten tasks from DeepMind Control Suite can be found at the official repository on GitHub, except Hopper Hop and Hopper Stand. We follow the official implementation of TD-MPC2 (Hansen et al., 2024), use the default hyperparameters, and select the default 5M parameters for the single task.

### E.4  SIMULUS

We follow the official implementation of Simulus (Cohen et al., 2025) and reproduce the results based on the default hyperparameters. We reproduce experiments on Atari 100K.

### E.5  DREAMERV3+RSSM-OP

Table 13: Game scores and human normalized mean score for DreamerV3+RSSM-OP on the 6 games from the Atari 100K.

| Game | EADream | DreamerV3+RSSM-OP | DreamerV3 |
|------|---------|-------------------|-----------|
| Assault | **883.4** | 792.5 | 723.6 |
| Breakout | **71.8** | 20.8 | 26.9 |
| Gopher | **4415.8** | 5287.3 | 4074.9 |
| Krull | **8729.6** | 7612.1 | 7796.6 |
| Ms Pacman | 1580.7 | 1429.3 | **1813.3** |
| Up N Down | **28408.2** | 20696.9 | 20183.2 |
| **Mean** ($\uparrow$) | **2.501** | 1.951 | 1.901 |

# F HYPERPARAMETER ANALYSIS

In terms of practical effort, tuning the additional hyperparameters required only a small number of validation runs on 6 Atari games that we used in our ablation studies: *Assault, Breakout, Gopher, Krull, Ms Pacman, and Up N Down*. Compared to previous work, such as DreamerV3 (Hafner et al., 2025), which tunes the base world model across many hyperparameter settings, this cost is relatively minor. In our experiment, once a reasonable default for the hyperparameters is chosen on these 6 Atari games, the values of the hyperparameters transfer well to tasks across DMC, Craftax, and DMC-GB2.

We posit that a universally applicable configuration exists for diverse scenarios. We use the default values of $C_I = 16$ and $\omega = 0.5$ for all experiments across benchmarks. As shown in Tables 14 and 15, our EADream consistently outperforms DreamerV3 even if the hyperparameters introduced by our method change significantly.

Table 14: Game scores and human normalized mean score for different values of $C_I$ on the 6 games from the Atari 100K.

| Game | EADream ($C_I = 8$) | EADream ($C_I = 16$) | EADream ($C_I = 32$) | DreamerV3 |
|---|---|---|---|---|
| Assault | 781.5 | **883.4** | 821.1 | 723.6 |
| Breakout | 43.2 | **71.8** | 23.1 | 26.9 |
| Gopher | 2997.1 | **4415.8** | 3316.5 | 4074.9 |
| Krull | 7043.0 | **8729.6** | 7575.8 | 7796.6 |
| Ms Pacman | 1568.6 | 1580.7 | 1566.5 | 1813.3 |
| Up N Down | 38616.1 | 28408.2 | **42507.2** | 20183.2 |
| **Mean** ($\uparrow$) | 2.082 | **2.501** | 2.144 | 1.901 |

Table 15: Game scores and human normalized mean score for different values of $\omega$ on the 6 games from the Atari 100K.

| Game | EADream ($\omega = 0$) | EADream ($\omega = 0.5$) | EADream ($\omega = 1$) | DreamerV3 |
|---|---|---|---|---|
| Assault | 942.8 | **883.4** | 639.1 | 723.6 |
| Breakout | 49.2 | **71.8** | 16.3 | 26.9 |
| Gopher | 3741.1 | **4415.8** | 2118.7 | 4074.9 |
| Krull | 8244.2 | 8729.6 | **8759.5** | 7796.6 |
| Ms Pacman | 1751.3 | 1580.7 | **2027.8** | **1813.3** |
| Up N Down | 19499.0 | 28408.2 | **37913.0** | 20183.2 |
| **Mean** ($\uparrow$) | 2.132 | **2.501** | 2.081 | 1.901 |

# G    Atari 1M

In this section, we investigate data scaling properties of the EAWM by increasing the number of interactions from 100K to 1M. To our surprise, our EADream performs much better than DreamerV3, as shown in Table 16. This result demonstrates that EAWM could be beyond the sample-efficient regime.

Table 16: Game scores and human normalized aggregate metrics on the Atari 1M. We highlight the highest and the second highest scores among all baselines in bold and with underscores, respectively. The results of DreamerV3(1M) are obtained from the official data of DreamerV3 (Hafner et al., 2025), where the agent was trained with a budget of 200M environment steps.

| Game | Random | Human | Ours(100K) | DreamerV3(1M) | Ours(1M) |
|---|---|---|---|---|---|
| Alien | 227.8 | 7127.7 | 776.4 | **1978.2** | 1550.2 |
| Amidar | 5.8 | 1719.5 | 144.2 | 265.5 | **672.8** |
| Assault | 222.4 | 742.0 | 883.4 | 1949.1 | **3369.2** |
| Asterix | 210.0 | 8503.3 | 1096.9 | 2917.3 | **6605.5** |
| BankHeist | 14.2 | 753.1 | 742.6 | 569.5 | **1272.5** |
| BattleZone | 2360.0 | 37187.5 | 13372.0 | 5261.8 | **39070.0** |
| Boxing | 0.1 | 12.1 | 85.4 | **96.6** | 93.6 |
| Breakout | 1.7 | 30.5 | 71.8 | 25.1 | **416.8** |
| ChopperCommand | 811.0 | 7387.8 | 904.0 | **4442.8** | 647.0 |
| CrazyClimber | 10780.5 | 35829.4 | 89038.6 | 108468.5 | **112524.0** |
| DemonAttack | 152.1 | 1971.0 | 152.2 | 644.4 | **2323.8** |
| Freeway | 0.0 | 29.6 | 0.0 | 7.9 | **34.0** |
| Frostbite | 65.2 | 4334.7 | 692.6 | 3741.9 | **3989.1** |
| Gopher | 257.6 | 2412.5 | 4415.8 | 14935.7 | **83756.6** |
| Hero | 1027.0 | 30826.4 | 8801.8 | 13600.5 | **20564.1** |
| Jamesbond | 29.0 | 302.8 | 337.2 | 465.6 | **688.5** |
| Kangaroo | 52.0 | 3035.0 | 3875.6 | **5432.7** | 5079.0 |
| Krull | 1598.0 | 2665.5 | 8729.6 | **13389.3** | 11534.0 |
| KungFuMaster | 258.5 | 22736.3 | 23434.6 | 35395.8 | **46526.0** |
| MsPacman | 307.3 | 6951.6 | 1580.7 | 3165.6 | **3449.9** |
| Pong | -20.7 | 14.6 | 20.1 | 9.4 | **21.0** |
| PrivateEye | 24.9 | 69571.3 | -472.5 | 882.4 | **5595.0** |
| Qbert | 163.9 | 13455.0 | 1664.4 | 5006.7 | **14411.0** |
| RoadRunner | 11.5 | 7845.0 | 12518.6 | **21404.0** | 19359.0 |
| Seaquest | 68.4 | 42054.7 | 557.9 | 1580.6 | **2387.8** |
| UpNDown | 533.4 | 11693.2 | 28408.2 | 115045.3 | **416219.0** |
| #Superhuman(↑) | 0 | N/A | 12 | 10 | **17** |
| Mean(↑) | 0.0 | 1.000 | 1.290 | 2.213 | **5.273** |
| Median(↑) | 0.0 | 1.000 | 0.651 | 0.782 | **1.188** |

# H EXTENDED RESULTS ON DEEPMIND CONTROL

For a more comprehensive evaluation of EAWM, we conducted an extensive experiment on all 20 tasks from the DeepMind Control Suite. As demonstrated in Table 17, EAWM has set a state-of-the-art result on the DeepMind Control Suite. Moreover, EAWM achieves the highest scores on half of the tasks among the baselines and performs consistently well.

Table 17: Scores achieved for all 20 tasks from DeepMind Control Suite with a budget of 500K interactions. We highlight the highest and the second highest scores among all baselines in bold and with underscores, respectively.

| Task | CURL | DrQ-v2 | DreamerV3 | TD-MPC2 | EADream (Ours) |
|---|---|---|---|---|---|
| Acrobot Swingup | 5.1 | 128.4 | 210.0 | 241.3 | **452.1** |
| Cartpole Balance | 979.0 | 991.5 | 996.4 | 993.0 | **999.4** |
| Cartpole Balance Sparse | 981.0 | 996.2 | **1000.0** | 1000.0 | 1000.0 |
| Cartpole Swingup | 762.7 | 858.9 | 819.1 | 831.0 | **871.4** |
| Cartpole Swingup Sparse | 236.2 | 706.9 | 792.9 | 790.0 | **825.8** |
| Cheetah Run | 474.3 | 691.0 | 728.7 | 537.3 | **874.3** |
| Cup Catch | 965.5 | 931.8 | 957.1 | 917.5 | **966.9** |
| Finger Spin | 877.1 | 846.7 | 818.5 | **984.9** | 596.7 |
| Finger Turn Easy | 338.0 | 448.4 | 787.7 | 820.8 | **916.6** |
| Finger Turn Hard | 215.6 | 220.0 | 810.8 | 865.6 | **921.5** |
| Hopper Hop | 152.5 | 189.9 | **369.6** | 267.6 | 311.5 |
| Hopper Stand | 786.8 | 893.0 | 900.6 | 790.3 | **926.2** |
| Pendulum Swingup | 376.4 | **839.7** | 806.3 | 832.6 | 835.0 |
| Quadruped Run | 141.5 | 407.0 | 352.3 | 283.1 | **648.7** |
| Quadruped Walk | 123.7 | **660.3** | 352.6 | 323.5 | 580.3 |
| Reacher Easy | 609.3 | 910.2 | 898.9 | **982.2** | 937.7 |
| Reacher Hard | 400.2 | 572.9 | 499.2 | **909.6** | 654.9 |
| Walker Run | 376.2 | 517.1 | 757.8 | 671.9 | **784.8** |
| Walker Stand | 463.5 | 974.1 | **976.7** | 878.1 | 966.6 |
| Walker Walk | 828.8 | 762.9 | **955.8** | 939.6 | 942.6 |
| Mean(↑) | 504.7 | 677.4 | 739.6 | 743.0 | **800.5** |
| Median(↑) | 431.8 | 734.9 | 808.5 | 831.8 | **872.8** |

# I    DMC-GB2

DMC-GB2 (Almuzairee et al., 2024) provides various test environments that are visually distinct from the training environment, as shown in Figure 7, and challenges RL agents to the ability of visual generalization. Algorithms such as SVEA (Hansen et al., 2021) and SADA (Almuzairee et al., 2024) are specifically designed for this setting and require paired original–augmented images. In contrast, EADream is applied to DMC-GB2 directly, without any modifications during migration from DeepMind Control, and is trained under fixed hyperparameters across five random seeds. Besides DreamerV3 (Hafner et al., 2025), we compare against DyMoDreamer (Zhang et al., 2025), a world model architecture that exploits temporal information via frame differences. As shown in Tables 18 to 20, EADream even slightly outperforms SADA, the state-of-the-art method tailored for this benchmark—an outcome that exceeds our expectations. We attribute this generalization ability to our Generic Event Segmentor (GES), which can capture event structures and filter out irrelevant events. Unlike purely reconstruction-driven objectives, which tend to overfit to pixel-level variations, GES focuses on changes that correspond to meaningful dynamics in the environment. This enables EAWM to disregard spurious fluctuations in the background—such as texture shifts, lighting changes, or irrelevant object motions—that do not alter the underlying task. By emphasizing task-relevant kinetic features, the model learns to prioritize interactions that matter for control, such as the movement of the agent's body or objects critical to task success. As a result, EAWM demonstrates robust visual generalization on DMC-GB2, suggesting strong potential for transfer to broader environments and real-world applications where background distractors are pervasive.

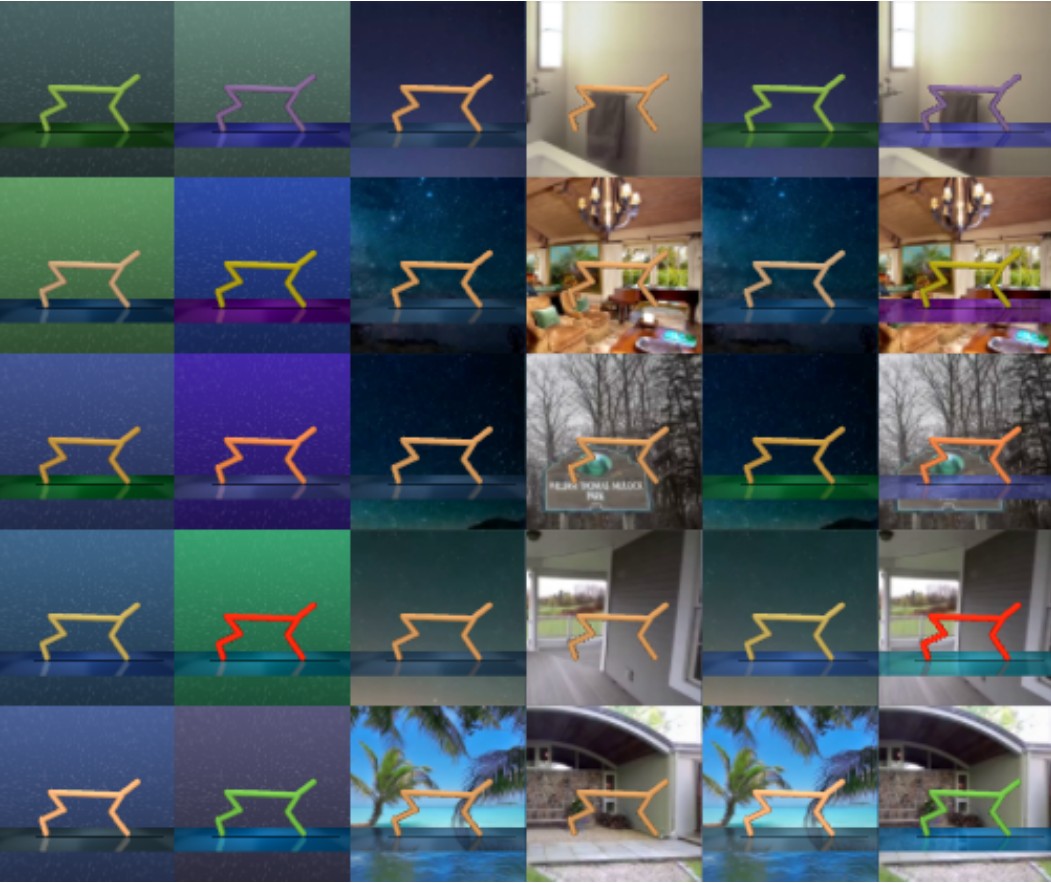

Figure 7. Snapshots of the Cheetah Run task in DMC-GB2 (Almuzairee et al., 2024). The test environments consist of Color Easy, Color Hard, Video Easy, Video Hard, Color Video Easy, and Color Video Hard (from left to right). Color refers to environments with randomized colors, while Video refers to the substitution of the original background for video from natural environments.

Table 18: Scores achieved in Color Hard test environments.

| Task | DrQ | SVEA | SADA | DreamerV3 | DyMoDreamer | EADream (Ours) |
|---|---|---|---|---|---|---|
| Cartpole Swingup | 441 | 478 | 716 | 432 | 495 | **774** |
| Cheetah Run | 178 | 133 | 239 | 417 | **593** | 567 |
| Cup Catch | 520 | 779 | **961** | 93 | 175 | 914 |
| Finger Spin | 466 | 802 | **868** | 218 | 681 | 576 |
| Walker Stand | 527 | 861 | 963 | 957 | **969** | 964 |
| Walker Walk | 265 | 667 | **825** | 617 | 766 | 705 |
| Mean(↑) | 400 | 620 | **762** | 456 | 613 | 750 |

Table 19: Scores achieved in Video Hard test environments.

| Task | DrQ | SVEA | SADA | DreamerV3 | DyMoDreamer | EADream (Ours) |
|---|---|---|---|---|---|---|
| Cartpole Swingup | 98 | 259 | 363 | 296 | 138 | **449** |
| Cheetah Run | 25 | 28 | 82 | 228 | 127 | **240** |
| Cup Catch | 111 | 416 | **662** | 53 | 92 | 288 |
| Finger Spin | 7 | 263 | **566** | 99 | 205 | 400 |
| Walker Stand | 154 | 429 | 702 | 816 | 809 | **872** |
| Walker Walk | 36 | 264 | 270 | 569 | **664** | 613 |
| Mean(↑) | 72 | 277 | 441 | 343 | 339 | **477** |

Table 20: Scores achieved in Color Video Hard environments.

| Task | DrQ | SVEA | SADA | DreamerV3 | DyMoDreamer | EADream (Ours) |
|---|---|---|---|---|---|---|
| Cartpole Swingup | 94 | 294 | 426 | 338 | 137 | **437** |
| Cheetah Run | 26 | 44 | 99 | 435 | 356 | **531** |
| Cup Catch | 122 | 484 | **697** | 16 | 87 | 573 |
| Finger Spin | 2 | 307 | **633** | 63 | 149 | 398 |
| Walker Stand | 170 | 659 | 906 | 946 | 837 | **952** |
| Walker Walk | 42 | 421 | 686 | 614 | **778** | 648 |
| Mean(↑) | 76 | 368 | 575 | 402 | 391 | **590** |

# J    ADDITIONAL RESULTS ON ATARI 100K

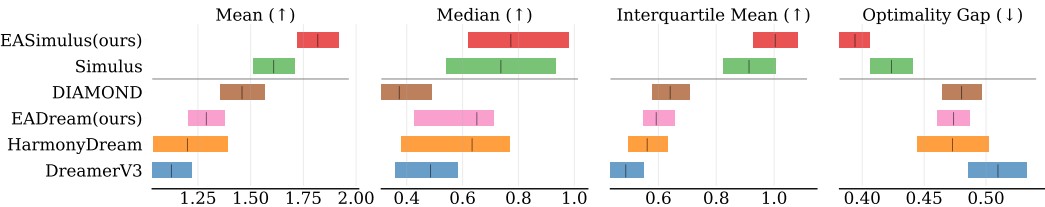

Figure 8. Mean, median, and inter-quantile mean (IQM) human-normalized scores and the optimality gap (Agarwal et al., 2021) with 95% stratified bootstrap confidence intervals on the Atari 100K benchmark.

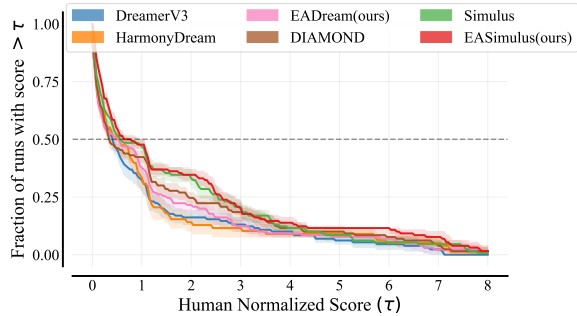

Figure 9. Performance profiles (Agarwal et al., 2021). The curve of each algorithm shows the proportion of runs in which human-normalized scores are greater than the given score threshold.

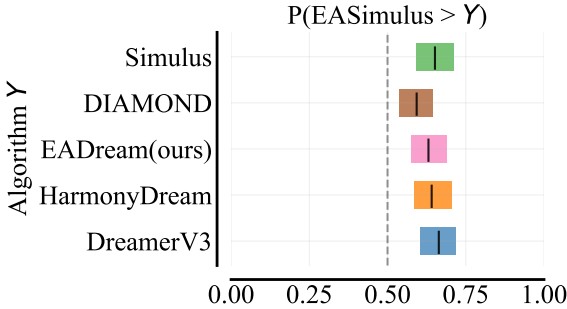

Figure 10. Each row represents the probability of improvement (Agarwal et al., 2021) that our algorithm outperforms the corresponding baseline in a randomly selected task from all tasks with 95% stratified bootstrap confidence intervals.

# K  ATARI 100K CURVES

## K.1  EADREAM

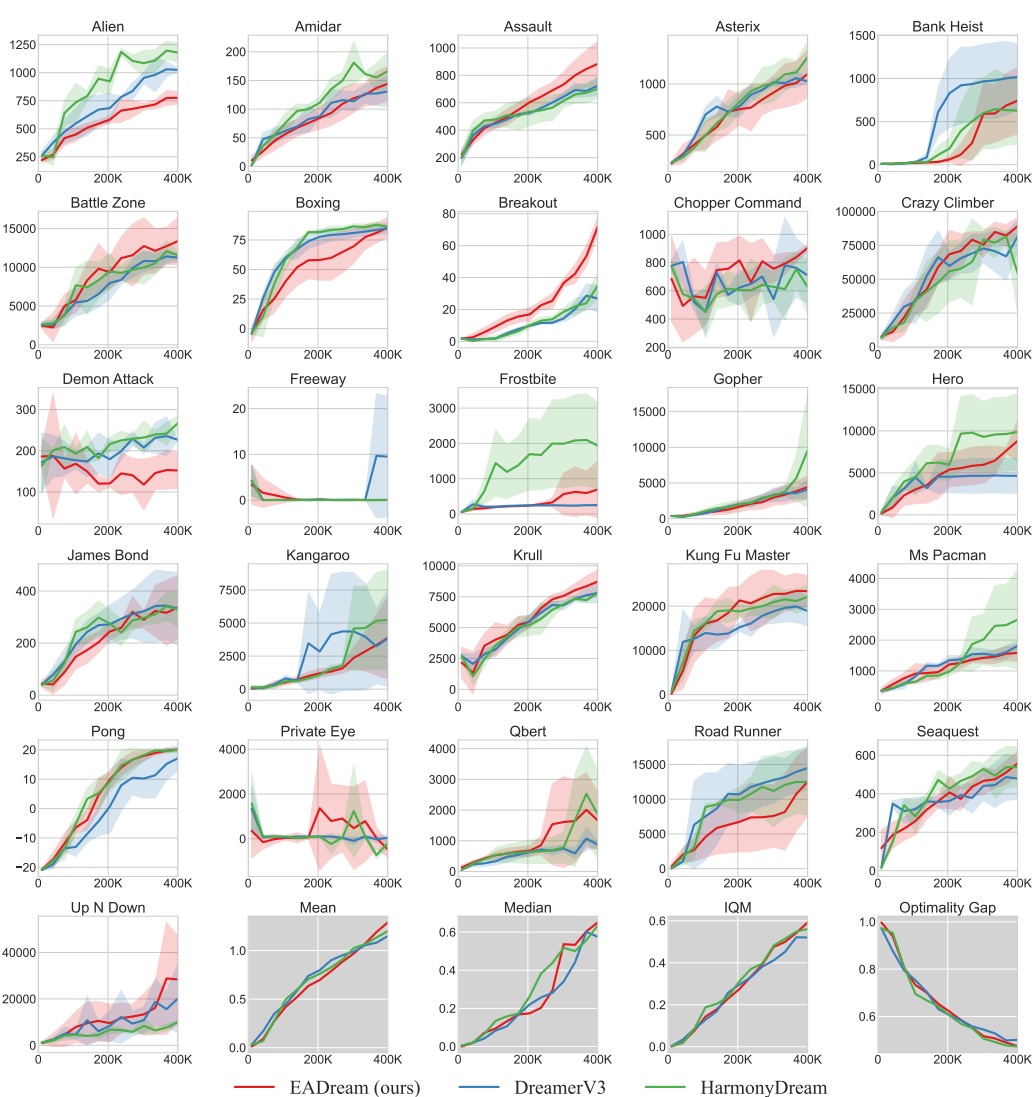

Figure 11. Training curves of EADream, DreamerV3, and HarmonyDream on the Atari 100K benchmark. 100K interaction data amounts to 400K frames.

## K.2 EASIMULUS

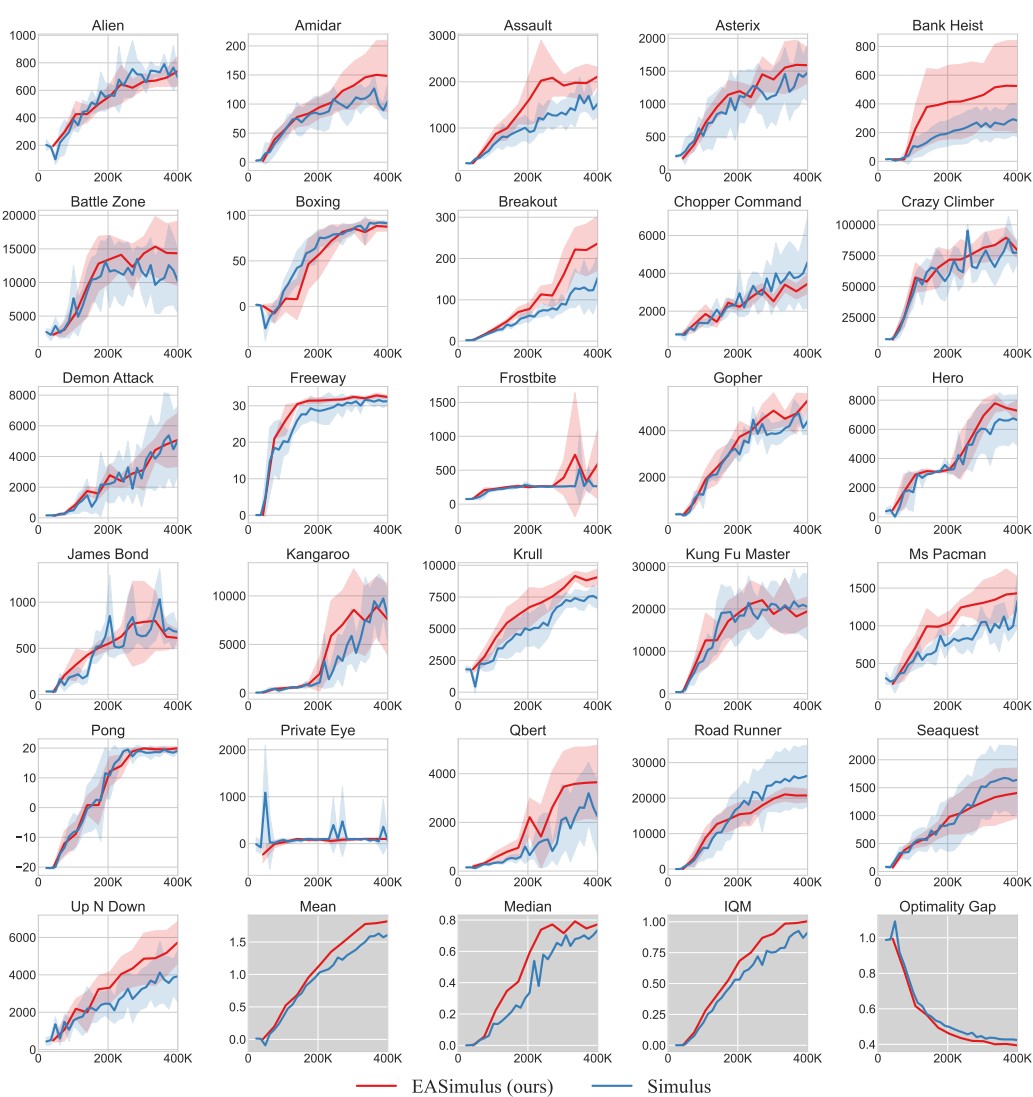

Figure 12. Training curves of EASimulus and Simulus on the Atari 100K benchmark. 100K interaction data amounts to 400K frames.

## L    DEEPMIND CONTROL 500K CURVES

Figure 13. Training curves of EADream and TD-MPC2 on the challenging tasks from DeepMind Control. 1M frames corresponds to 500K interaction data.

## M    COMPUTATIONAL RESOURCES

**EADream** Experiments of EADream on Atari 100K were conducted with NVIDIA V100 32GB GPUs. Training on Atari 100K, with three tasks running on the same GPU in parallel, took about 1.2 days, resulting in an average of 0.4 days per environment (+11% more than DreamerV3). Experiments on DMC were conducted with NVIDIA GeForce RTX 4090 24GB GPUs. Training on DMC, with three tasks running on the same GPU in parallel, took 1.8 days, resulting in an average of 0.6 days per environment (+13% more than DreamerV3). On V100 GPUs, training a baseline DreamerV3 on DMC-GB2 cost us 2.3 days, while training an EADream model cost us 2.6 days.

**EASimulus** Our experimental evaluations of Simulus were finished on NVIDIA GeForce RTX 4090 24GB GPUs. Training a baseline Simulus model on Atari 100K and Craftax 1M required approximately 0.7 day and 6 days per task, respectively. Compared to the training cost of our EASimulus (on Atari 100K and Craftax 1M, approximately 0.8 day and 7 days per task, respectively).

## N    VISUALIZATIONS

### N.1    ATARI 100K

We visualize the predictions of observations by our EAWM compared to pretrained video generation models. Since existing models were not pre-trained with low-resolution images, we resize images to $512 \times 512$ as inputs for pre-trained video generation models. We tried several pre-trained video generation models (Blattmann et al., 2023; Esser et al., 2024) to generate future frames conditioned on the past frames and proper prompts, among which the best results are selected. As shown in Figure 14 and Figure 15, the pre-trained model often misses the moving patterns of small targets, while our EAWM can make fine-grained predictions of future frames.

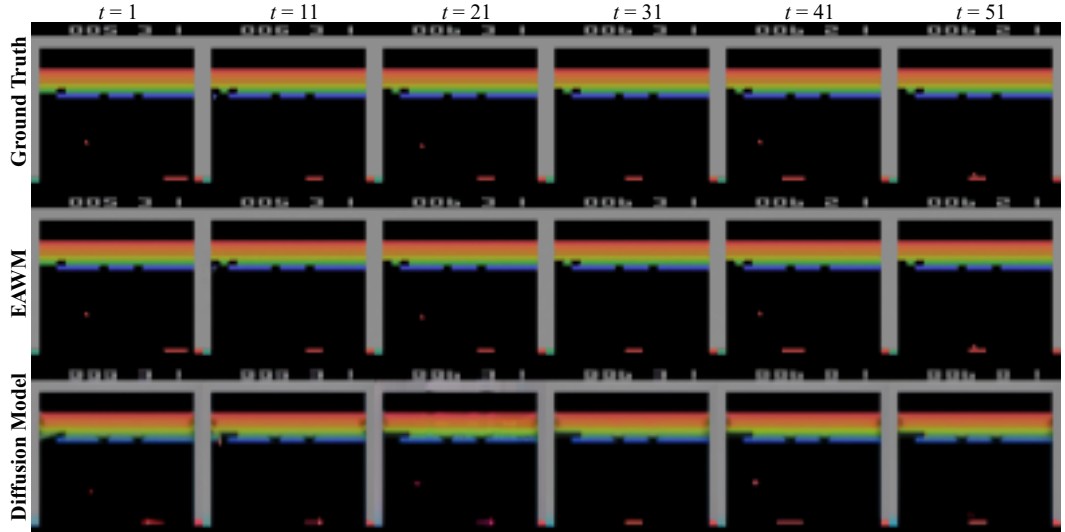

Figure 14. Comparison of predicted frames for the game Breakout by our EAWM and pretrained diffusion models. Notably, at time $t = 11$, EAWM succeeds in predicting the change of score from 5 to 6 in the upper part of the frame and the color change of the tiny ball. However, the pre-trained Stable Diffusion model even misses information about the tiny ball at time $t = 51$.

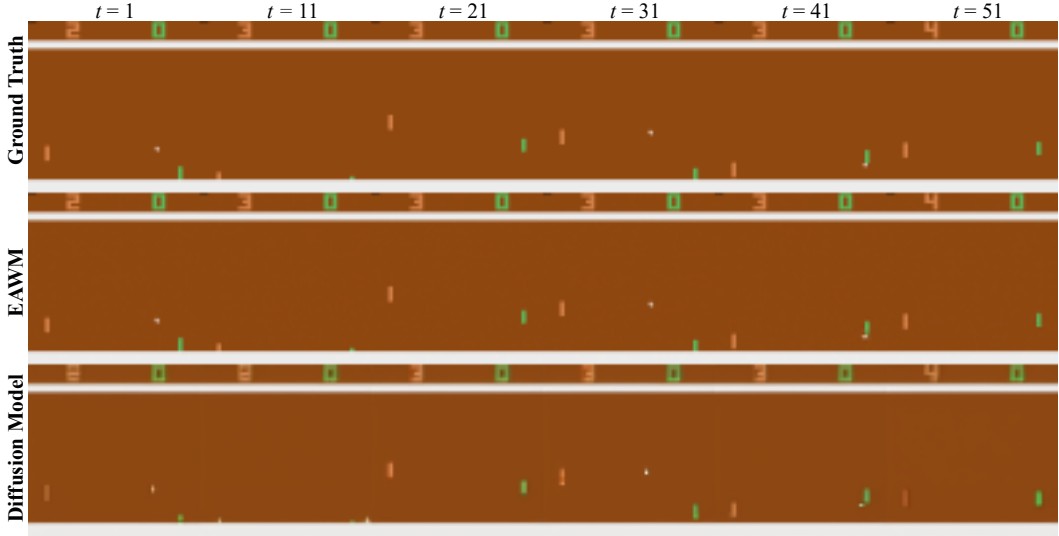

Figure 15. Comparison of predicted frames for the game Pong by EAWM and pretrained diffusion models. EAWM succeeds in predicting the change of score from 3 to 4 in the upper part of the frame at time $t = 11$ and has a more accurate estimation of the moving tiny objects than the pre-trained diffusion model.

### N.2 DMC-GB2

We provide empirical evidence to support that observation prediction is much harder to generalize to unseen environments than event prediction on DMC-GB2 (Almuzairee et al., 2024). As shown in Figures 16 to 18, EAWM, DreamerV3 (Hafner et al., 2023), and DyMoDreamer (Zhang et al., 2025) make satisfactory future predictions during training. However, the observation prediction by these models fails in the unseen test environments. We observe that event prediction generalizes well to unseen test environments where the background may be changing continuously. In addition, these results provide a reasonable rationale for the excellent performance of EADream on DMC-GB2.

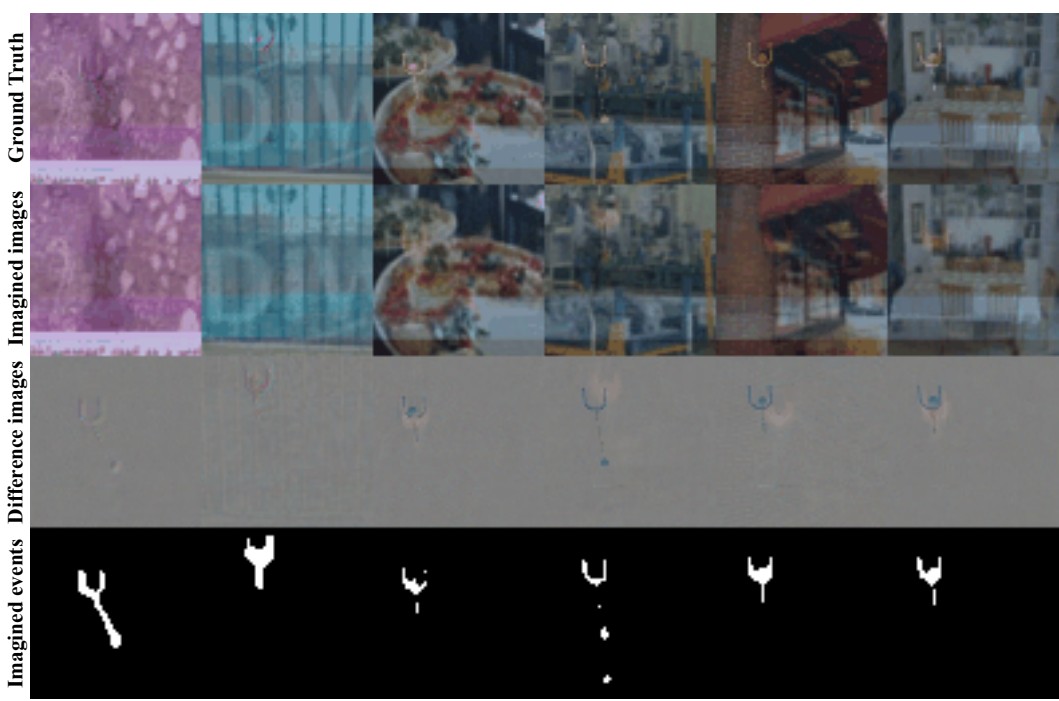

Figure 16. Predictions in training environments of *Cup Catch* by EADream at imagination step 9.

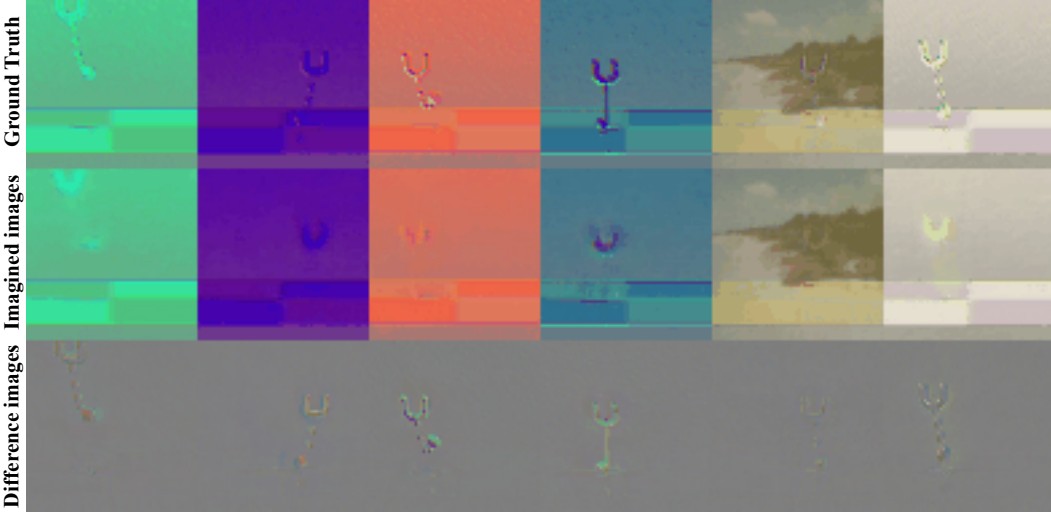

Figure 17. Predictions in training environments of *Cup Catch* by DreamerV3 at imagination step 9.

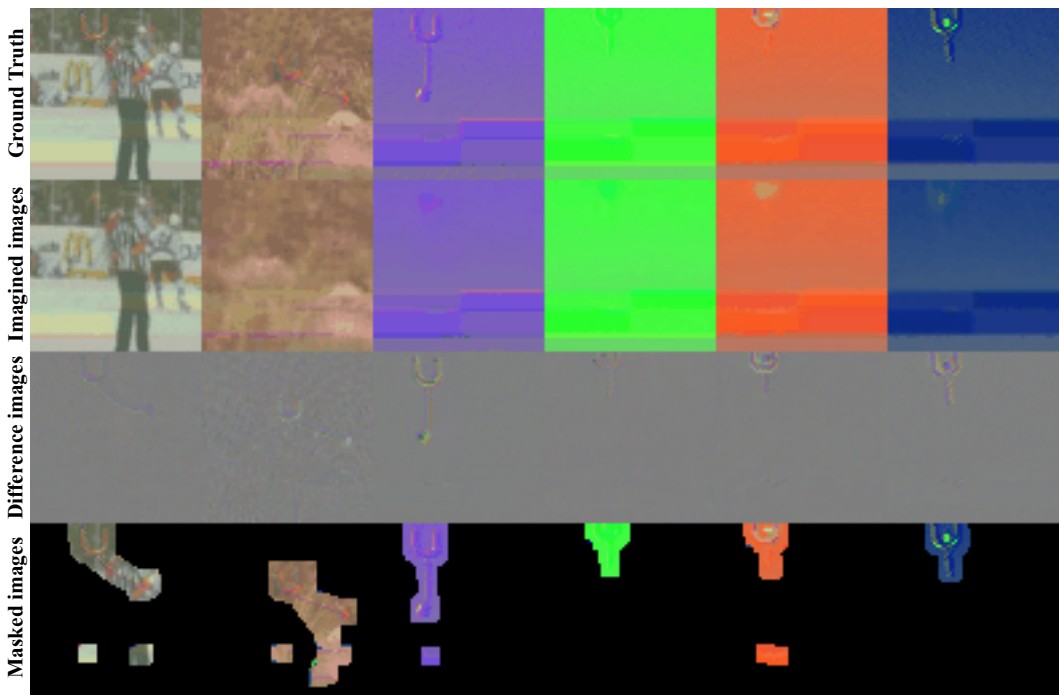

Figure 18. Predictions in training environments of *Cup Catch* by DyMoDreamer at imagination step 9. Given a predefined threshold $\epsilon$, DyMoDreamer defines a mask $\mathbf{M}_t$ for the current observation $\mathbf{o}_t$, where each element $M_{t,h,w,c} = 1$ if $\|o_{t,h,w,c} - o_{t-1,h,w,c}\|_2 > \epsilon$. Then, masked images are generated via $\mathbf{M}_t \cdot \mathbf{o}_t$ and encoded to obtain dynamic features.

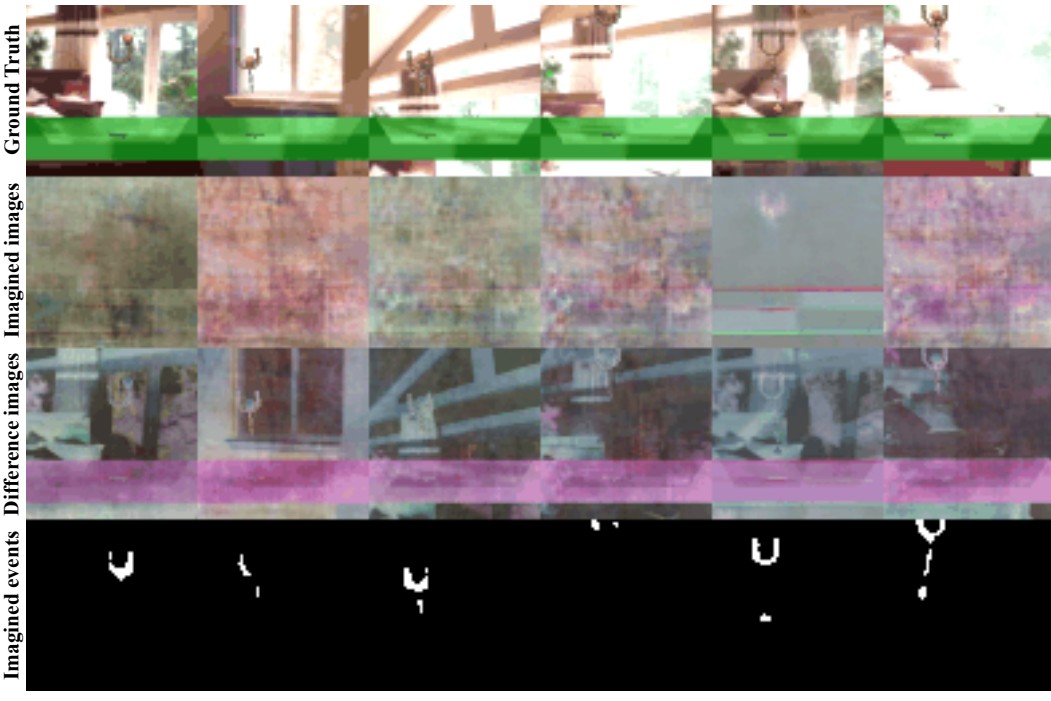

Figure 19. Predictions in unseen test environments of *Cup Catch* by EADream at imagination step 9.

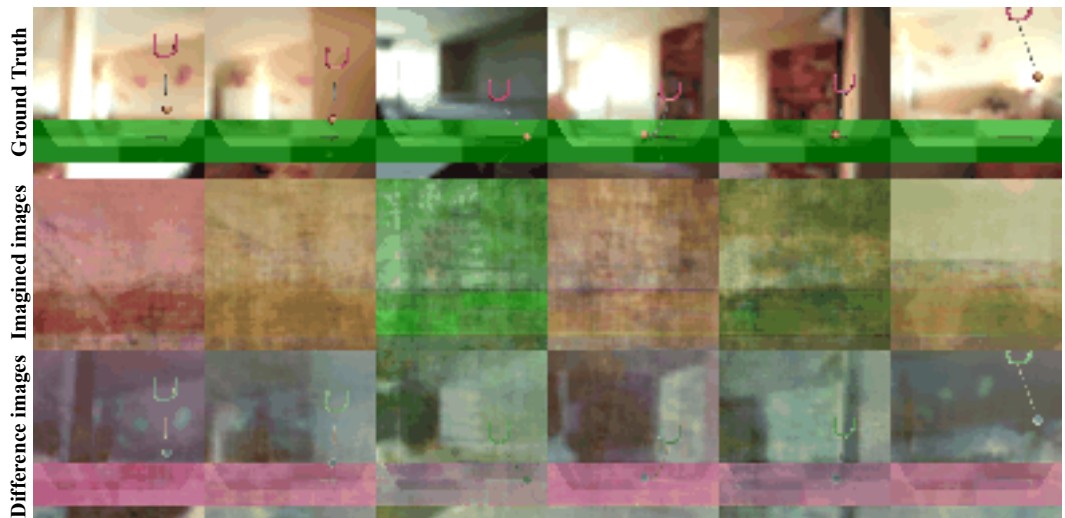

Figure 20. Predictions in unseen test environments of *Cup Catch* by DreamerV3 at imagination step 9.

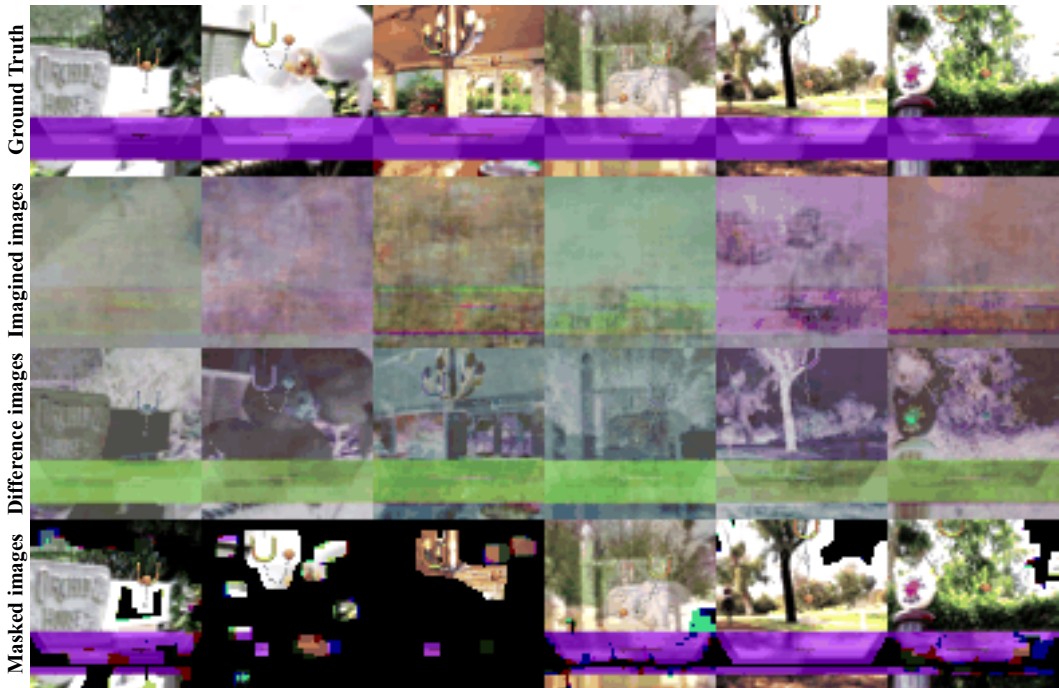

Figure 21. Predictions in unseen test environments of *Cup Catch* by DyMoDreamer at imagination step 9. Masked images proposed by DyMoDreamer fail to produce sparse dynamic features in environments where the background is changing.

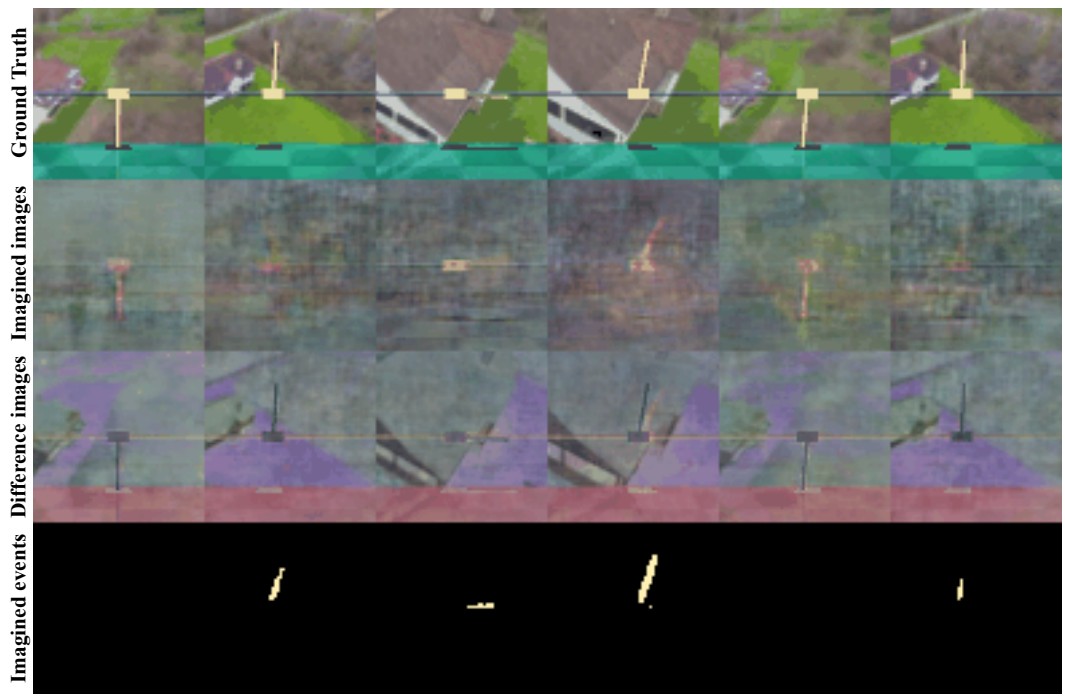

Figure 22. Predictions in unseen test environments of *Cartpole Swingup* by EADream at imagination step 3.

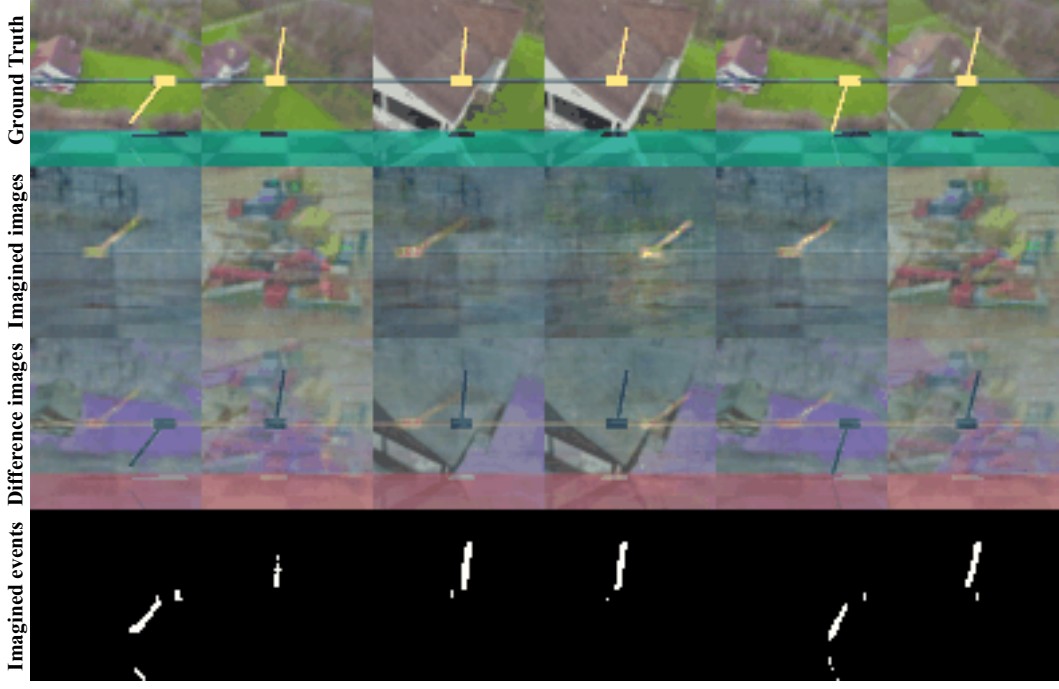

Figure 23. Predictions in unseen test environments of *Cartpole Swingup* by EADream at imagination step 33.

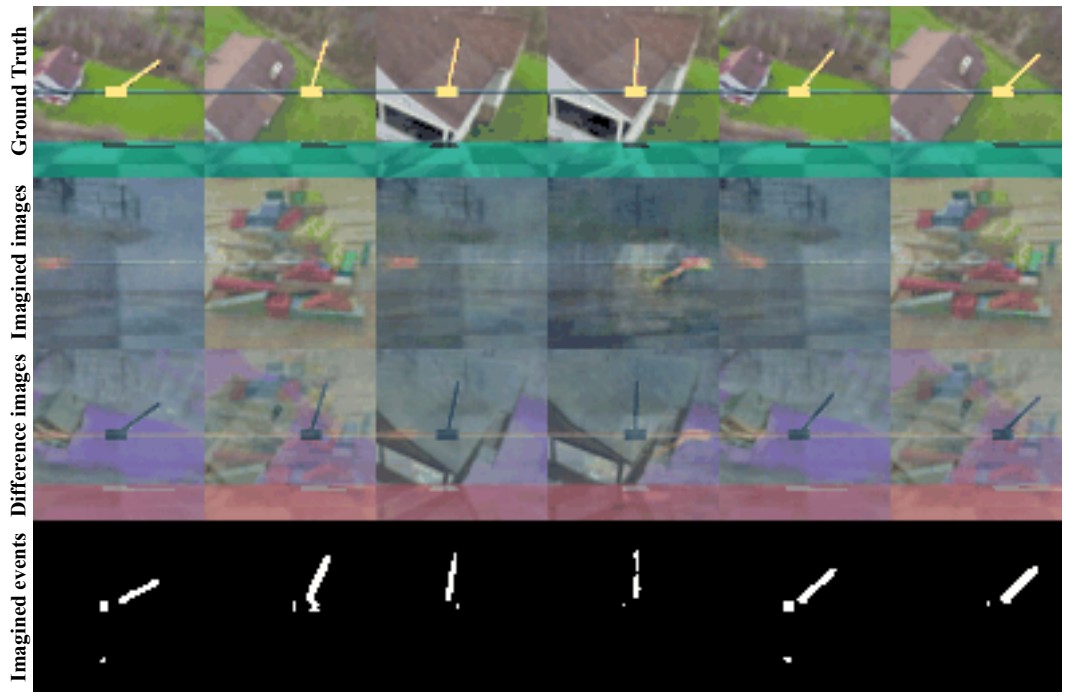

Figure 24. Predictions in test environments of *Cartpole Swingup* by EADream at imagination step 63.

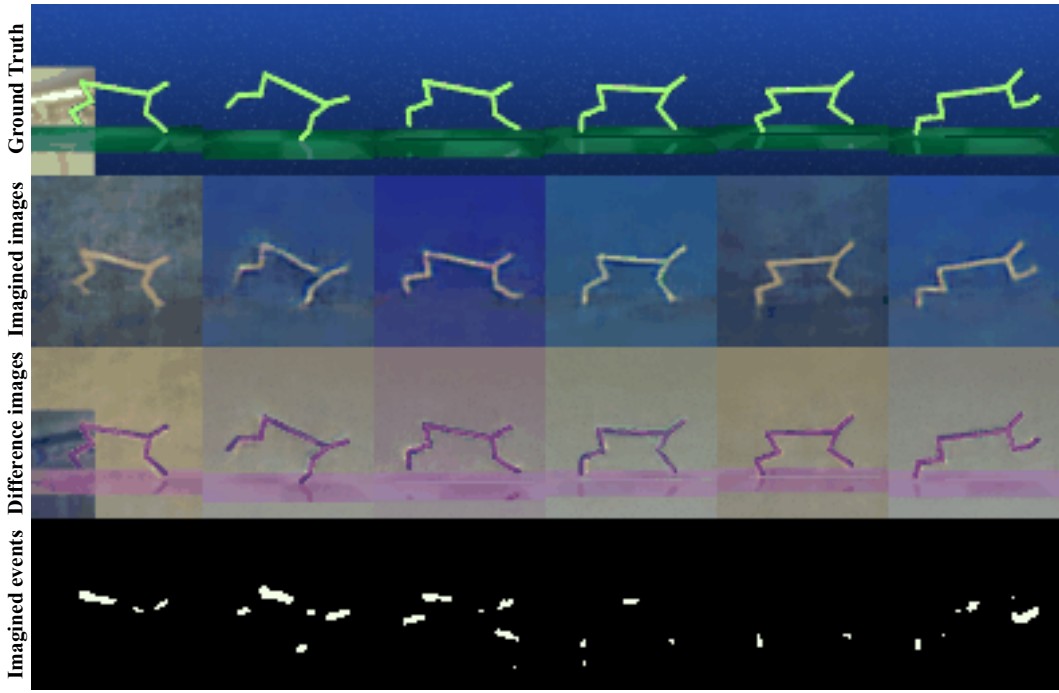

Figure 25. Predictions in test environments of *Cheetah Run* by EADream at imagination step 3.

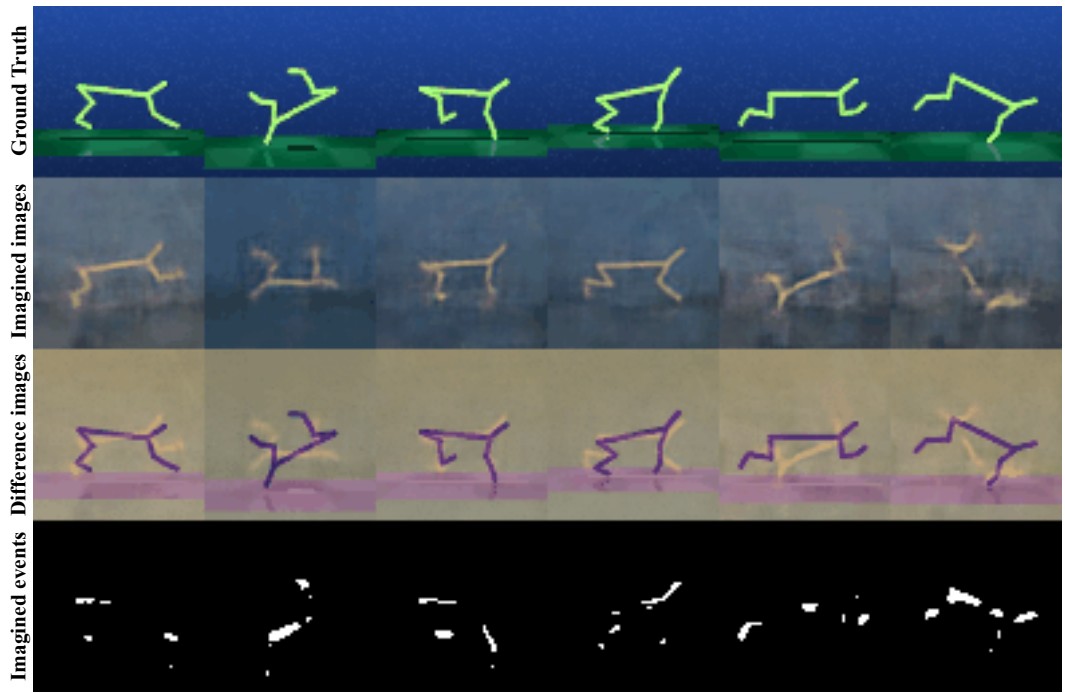

Figure 26. Predictions in unseen test environments of *Cheetah Run* by EADream at imagination step 33.

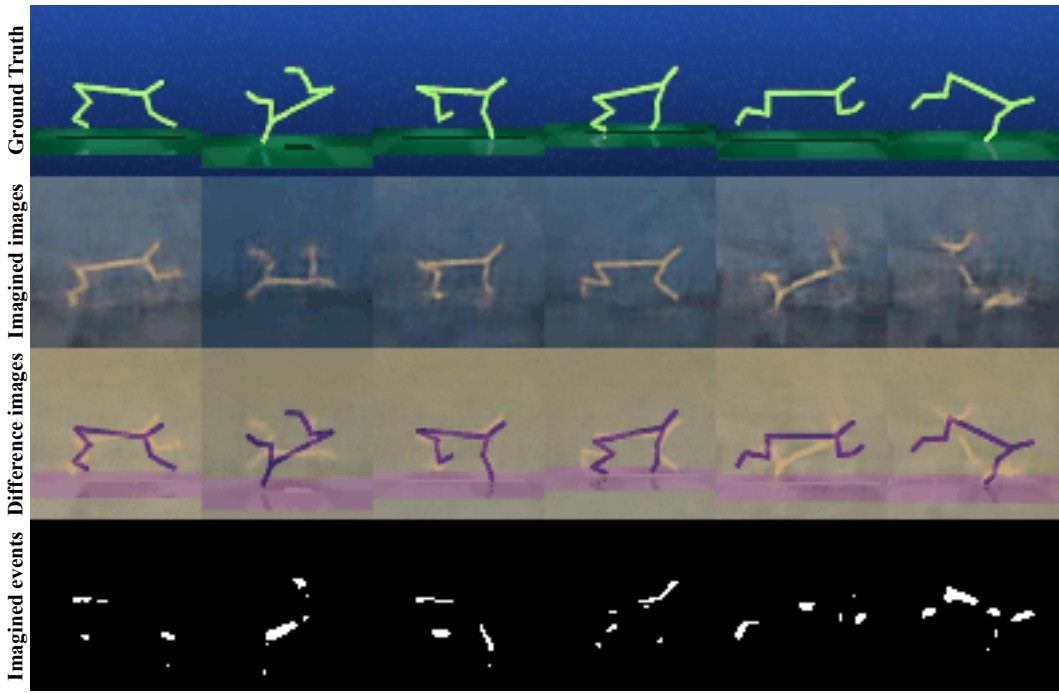

Figure 27. Predictions in unseen test environments of *Cheetah Run* by EADream at imagination step 63.

## O   USE OF LARGE LANGUAGE MODELS

This paper made limited use of large language models (LLMs) as general-purpose writing and editing assistants. Specifically, GPT-5-thinking was used to help refine the clarity, conciseness, and flow of certain sections and to polish technical descriptions for readability. All research ideas, experiments, analyses, and results presented in this paper were conceived, designed, and executed entirely by the authors without assistance from LLMs. The LLM did not contribute to the novelty of the method, experimental design, or interpretation of results, and was not used to generate figures, code, or data.

