# OpenReview forum: "From Observations to Events: Event-Aware World Models for Reinforcement Learning"
_ICLR.cc/2026/Conference — ICLR 2026 Poster_

### Official Review · Reviewer_6jd5 · 2025-10-30

**Soundness:** 2
**Presentation:** 1
**Contribution:** 3
**Rating:** 2
**Confidence:** 4

**Summary:**

The paper adds an "event-aware" layer to world models so that agents learn from meaningful changes (events) rather than raw frames alone. It defines events across modalities, predicts them with a dedicated head, and uses a Generic Event Segmentor (GES) to weight losses. Plugging this into DreamerV3 and Simulus yields strong empirical gains across Atari 100K, Craftax, DMC, and DMC-GB2.

**Strengths:**

- The paper proposes a novel extension that can be attached to existing world models.
- Strong empirical performance across benchmarks.

**Weaknesses:**

- Event definition in Section 2.2 feels rushed:
  - Eq. (1): the image intensity $I_t$ is not explicitly introduced in the text.
  - Line 151: "direction of the event" is unclear, does this mean the sign of the brightness change?
  - For ordinal data, the event is defined as $p_i$. For consistency with visual data it should likely be a tuple e_{t,i}.
  - Eq. (2): $\Delta o_i$ is not defined.
  - This carries into Eqs. (3)-(4): wwhat exactly are the events $e$? In places it seems $p$ might be intended instead of $e$.
- Discrepancies in Figure 3 vs. Eq. (3):
  - The sequence model is labeled $F_\theta$ in the figure but $f_\theta$ in the equation.
  - In Eq. (3), the representation model is conditioned on $y_t$, in the figure it is conditioned on $h_t$.
  - In Eq. (3), the event predictor conditions on $y_{t-1}, y_t$, in the figure it appears to use $y_t, y_{t+1}$.
  - Why is the observation predictor categorized under "EA" rather than "WM"?
- The sentence "We employ Adaptive Gaussian Mixture Models to automatically detect such events from video" needs concrete detail. What features are modeled, how thresholds are chosen/updated, and how detections map to the event tuple. This is central to the method but currently vague.
- The description of the Geneirc Event Segmentor is hard to follow:
  - How is $\alpha_t$ computed exactly, do indices $i$ range over pixels or batches? This should be made precise.
  - Line 272: please define "event boundaries."
  - The role of GES (as a weighting coefficient for the event/observation losses) emerges only from the equations. The text says "reallocate attention" but should explicitly tie to Eqs. (7)-(8).
  - The function $g$ in Eq. (6) appears without explanation in the main text, only the appendix clarifies it.
  - From Eq. (6), GES seems independent of $\theta$. If it has no learned parameters, why is it part of the world model rather than a fixed weighting step?
- Reported scores show discrepancies vs prior work:
  - DreamerV3 numbers on Atari 100K are lower than the original paper (mean 1.15 reported here vs 1.25), which makes the gap to EADream's 1.29 smaller than suggested. Please clarify whether these were re-runs and why they differ.
  - For Simulus, the reported mean and median (1.61 and 0.74) differ from the Simulus paper (1.65 and 0.98). Please explain the deviation.
- Ablations are unclear:
  - "No Event Predictor": What exactly is disabled? If event prediction is off, shoudn't the model reduce to the original world model? Why do "EADream w/o Event Predictor" and DreamerV3 differ? Is this related to the RSSM-OP change, but if so, why is there the same effect for Simulus?
  - "No GES": If GES is off, is $\omega$ also set to $0$? If so, consider an additional ablation where GES is on but $\omega = 0$ to disentangle effects.
  - "Without Observation Prediction": RSSM-OP seems to matter a lot. An ablation "Dreamer + RSSM-OP" would isolate its contribution.
- Minor: Missing citations to Craftax and DMC in Section 4.1.

I'm willing to raise my score if the presentation is improved and these points are addressed, as the method and results look very promising. My current score is low because, in its present form, the paper should not be accepted due to several presentation issues, but if these are fixed I would lean toward acceptance.

**Questions:**

- The method adds several hyperparameters on top of already hyperparameter-heavy world models. How sensitive are the new hyperparameters in practice, and how difficult was tuning?
- Please address the clarification requests in the weaknesses above.

---

> ### Author Response · Authors · 2025-11-26
> **Rebuttal by Authors**
>
> We sincerely thank the reviewer for acknowledging our experimental results and for the constructive feedback on the presentation and the ablation study. We have addressed all the points raised in detail below.
>
> ---
>
> > **Weakness 1:** Event definition in Section 2.2 feels rushed:
>
> **Response:**
> - We apologize for the confusion caused by the undefined $I\_t$. We have included the definition for the symbol $I\_t(x\_i,y\_i)$ in Lines 146-148.
> - You are right. The sign of the brightness change denotes the direction of the event. We have clarified this in Lines 154-155.
> - Thank you for the valuable suggestion. To provide a clearer presentation, we have revised the symbols of events of each modality with different superscriptions as follows: $e^\\text{vis}\_{t, i}$ for visual inputs, $e^\\text{ord}\_{t,i}$ for ordinal data, $e^\\text{nom}\_{t,i}$ for nominal data.
> - $\\Delta o\_i$ represents the temporal difference between the adjacent ordinal data.  That is, $\\Delta o\_i = o\_t(i) - o\_{t-1}(i)$. To further improve the readability, we have substituted $(o\_t(i)-o\_{t-1}(i))$ for $\\Delta o\_i$ in Equation (2), where $o\_t(i)$ is the value of the $i$-th ordinal data at time $t$.
> -  In Eq. (3), we have used $ p\_{\theta}(\mathbf{\hat{e}}\_t|\cdot)$ to denote the distribution of multimodal-events for general purposes and $\mathbf{\hat{e}}\_t$ to denote events sampled from the distribution. We agree with you that it should be $p^{(m)}\_i$ when computing loss of events for the $m$-th modality in Eq. (4). We have corrected this formula.
>
> These changes above have been highlighted $\\color{blue}{\\text{in blue}}$ in Lines 145-192 and Lines 279-281 of the revised manuscript.
>
> > **Weakness 2:** Discrepancies in Figure 3 vs. Eq. (3).
>
> **Response:**
>
> - Thank you for pointing out the erratum. We have revised the label in Eq. (3) to keep it consistent with Figure 3.
> - The representation model should be conditioned on $\\mathbf{y}\_t$. We have corrected it.
> - We apologize for the inconsistency in the start time between Eq. (3) and Figure 3. Since the output of the sequence model $\\mathbf{y}\_t$ contains information derived from $\\mathbf{o}\_{t-1}$, we have adjusted Eq. (3) accordingly to maintain temporal consistency across formulations.
> - We agree with you that the observation prediction should belong to "WM", and we have corrected this in the revised manuscript. In addition, we have included the Generic Event Segmentor (GES) in Figure 3 for better presentation.
>
>
> > **Weakness 3:** The sentence "We employ Adaptive Gaussian Mixture Models to automatically detect such events from video" needs concrete detail. What features are modeled, how thresholds are chosen/updated, and how detections map to the event tuple. This is central to the method but currently vague.
>
> **Response:**
> That's a good point. We have elaborated the process of capturing events with Adaptive Gaussian Mixture Models (AGMMs) in the revised manuscript (Lines 232-253):
>
> For visual inputs, instead of using the primitive definition of events, we employ Adaptive Gaussian Mixture Models (AGMMs) (Zivkovic & Van Der Heijden, 2006) to reduce false alarms from noise or gradual changes in brightness. That is, we treat the event as a statistically significant deviation from a learned multi-modal distribution of each pixel. The model maintains a mixture of $K$ Gaussian components for each pixel:
> $$
>     p(L\_t(x\_i,y\_i))
>     =\\sum\_{k=1}^{K} w\_{k,t,i} \,
>     \\mathcal{N}\big(L\_t(x\_i,y\_i); \\mu\_{k,t,i}, \\Sigma\_{k,t,i}\big),$$
>
> where $w\_{k,t,i} \\ge 0$ are the mixture weights with $\\sum\_{k=1}^{K} w\_{k,t,i} = 1$,
> ${\\mu}\_{k,t,i}$ are the component means, and $\\Sigma\_{k,t,i}$ are the
> covariance matrices. We compute the squared Mahalanobis distance $D\_{k,i}$ between the next observation $L\_{t+1}(x\_i,y\_i)$  and the component means $\\mu\_{k,t,i}$ for each component $k$:
>
> $$D\_{k,i}=(L\_{t+1}(x\_i,y\_i) - \\mu\_{k,t,i})^\\top
> \\Sigma_{k,t,i}^{-1}
> (L\_{t+1}(x\_i,y\_i) - \\mu\_{k,t,i}).
> $$
> If $D\_{k,i}$ is larger than the threshold $C\_I$, we generate an event $e^\\text{vis}\_{t+1,i}=(x\_i,y\_i,t+1,p\_i)$ on that pixel and a new component will be generated; otherwise, $L\_{t+1}(x\_i,y\_i)$ matches a Gaussian component, say $k$, and the weight of component $k$ will be increased. We also generate an event $e^\\text{vis}\_{t+1,i}$ when the matched component exhibits low weight in comparison to other components. To summarize, events are generated at spatio-temporal locations where the model is either surprised (large $D\_{k,i}$) or uncertain (low weight), which aligns well with event-based systems (Muir & Sheik, 2025) that concentrate computation on informative changes rather than on every frame and every pixel. As a result, the event stream is sparse yet information-rich, and less sensitive to irrelevant noise and slow illumination changes than the raw observations.

---

> ### Author Response · Authors · 2025-11-26
> **Rebuttal by Authors**
>
> > **Weakness 4:** The description of the Geneirc Event Segmentor is hard to follow.
>
> **Response:**
> -  Thank you for the careful reading. For each modality $m$, we compute $\\alpha^{(m)}_t=\\sum\_{i=1}^{N^m}\\frac{1}{N^m}\\mathbb{I}(e\_{t,i}^{(m)}\\text{ occurs})$, where $N^m$ denotes the dimensionality (size) of the observation for the $m$-th modality. The index $i$ ranges over individual observation elements, e.g., pixels in the case of visual inputs. We have included the descriptions of $N^m$ and $\\alpha^{(m)}_t$ in Lines 282 and 296.
> - Event boundaries are defined as the start and end time of a meaningful slice of observation streams. We have improved the description in Lines 291-292 correspondingly.
> - We agree with you that the original description was vague. We have explicitly tied the role of GES to Eq. (7)-(8) and provided more details in Lines 302-313:
>
>     If $g(\\alpha^{(m)}\_t,\\alpha\_{\\text{thr}}^{(m)})=0$, we say that an event boundary is detected. In this case, event prediction is devoid of meaning and should be suppressed. Therefore, Equation (7) with GES turns into:
>     $$
>        \\mathcal{L}_{\text{e}}(\\theta)=\\sum\_{m=1}^{M}\\beta^{(m)}\_{\\text{e}}g(\\alpha^{(m)}\_t,\\alpha\_{\\text{thr}}^{(m)})\\mathcal{L}^{(m)}\_{\\text{e}}(\\theta).
>     $$
>     improve accuracy on informative parts of the observations, rather than uniformly over all observations. However, when GES detects an event boundary where the priority of dynamic events is not suitable, the world model should reallocate attention from events to raw observations and focus on modeling raw observations. We implement this via an event-aware observation loss:
>     $$
>         \\mathcal{L}\_{\text{o}}(\\theta)\\doteq \\mathcal{L}'\_{\\text{o}}(\\mathbf{o}\_{t},\\hat{\\mathbf{o}}\_{t})+
>         \\sum\_{m=1}^{M}\\sum\_{i=1}^{N^m}\\omega g(\\alpha^{(m)}\_t,\\alpha\_{\\text{thr}}^{(m)})\\left[\\mathbb{I}(e\_{t,i}^{(m)}  \\text{occurs})-1\\right]\\mathcal{L}'\_{\\text{o}}(o\_{t,i},\\hat{o}\_{t,i}),
>     $$
>     where $\\omega \\in [0,1]$ is the event-aware weight to balance attention of the overall observations and the part of the observations where events take place. $\\mathcal{L}'\_{\text{o}}$ denotes the loss function of observations given by base world models by default if it exists.
>
> - For DreamerV3, to achieve computational efficiency, we implement a simple form of GES:
>
>   $$g(\\alpha^{(m)}\_t,\\alpha\_{\\text{thr}}^{(m)})=\\mathbb{I}(\\alpha\_t^{(m)}<\\alpha\_{\\text{thr}}^{(m)}).$$
>
>   For EASimulus, the GES increases as events become sparser, allowing them to be better highlighted:
>   $$g(\\alpha^{(m)}\_t,\\alpha\_{\\text{thr}}^{(m)})=\\mathbb{I}(\\alpha\_t^{(m)}<\\alpha\_{\\text{thr}}^{(m)})/{\\text{arsinh}\\left[\\text{clip}\\left ( \\frac{\\alpha\_t^{(m)} }{\\alpha\_{\\text{thr}}^{(m)}},\\epsilon\_{\\alpha},1 \\right )\\right]},$$
>
>   where $\\epsilon_{\\alpha}> 0$ is a coefficient to smooth the curve of $\\alpha\_t^{(m)}$ and stablize the training process. We have detailed the function of $g$ in Section 4.2 "Implementation Details and Results".
>
> - Thank you for the close reading. You are correct that our current implementation of the GES does not introduce additional trainable parameters to enhance computational efficiency. Future work can parameterize and train the GES to detect event boundaries by using changes in the reconstruction error of video snippets. GES is a part of our proposed EAWM since it modulates the training of the WM part in Eq. (3). Our rationale for treating it as part of the EAWM is that it changes the effective training objective of the world model, and therefore the learned parameter $\theta^*$ of world models. Nevertheless, our EA part (including GES) is a training-time mechanism that shapes the representation of world models, but can be omitted at test time without changing the inference procedure. We have clarified this point in Lines 297-299.
>
> > **Weakness 5:** Reported scores show discrepancies vs prior work.
>
> **Response:**
> - The results on Atari 100k are based on the established re-implementation of the original version of DreamerV3 (Hafner et al., 2023) in PyTorch using the default hyperparameters over 5 seeds. In [the original paper](https://arxiv.org/pdf/2301.04104v1), they reported the mean HNS score of 112%. Our experiments of DreamerV3 were conducted on the same device (8 NVIDIA V100 32GB GPUs) and conda environment as EADream.
> - We follow the official implementation of Simulus (Cohen et al., 2025) and reproduce the results based on the suggested hyperparameters over 5 seeds. The experiment was conducted on the same device (8 NVIDIA 4090 24GB GPUs) and conda environment as EASimulus.
>
> We have explained the discrepancies in the title of Table 1 and Appendix E.

---

> ### Author Response · Authors · 2025-11-26
> **Rebuttal by Authors**
>
> > **Weakness 6:** Ablations are unclear.
>
> **Response:**
> - "No Event Predictor": We apologize for the confusion. In this case, the GES still influences the loss function in Equation (8). It is correct that RSSM-OP is also employed for EADream. We have clarified this point in the ablation studies.
> - "No GES": Yes, you'are right that event weights is 0 if the GES outputs 0. Follow your suggestion, we have conducted an ablation study where $\omega=0$ to disentangle effects. In addition, we have included an additional study where $\omega=1$ for hyperparamter analysis. Table 6 demonstrates that the event prediction loss with GES does increase the performance of EAWM.
>
> $$ \\textbf{Table 6. Game scores and human normalized mean score for different values of }\omega\textbf{ on the 6 games from the Atari 100K.}$$
> | Game  | EADream $(\\omega=0)$ | EADream $(\\omega=0.5)$ | EADream $(\\omega=1)$ | DreamerV3|
> |:-:|:-:|:-:|:-:|:-:|
> | Assault| 942.8| **883.4**|639.1| 723.6|
> | Breakout| 49.2| **71.8**|16.3| 26.9 |
> | Gopher | 3741.1               | **4415.8**        | 2118.7 | 4074.9|
> | Krull                      | 8244.2               | 8729.6                 | **8759.5**      | 7796.6          |
> | Ms Pacman                  | 1751.3               | 1580.7                 | **2027.8**      | 1813.3 |
> | Up N Down                  | 19499.0              | 28408.2                | **37913.0**     | 20183.2         |
> | **Mean** ($\\uparrow$) | 2.132                | **2.501**         | 2.081   | 1.901           |
> - "Without Observation Prediction": To isolate the contribution of RSSM-OP, we have conducted an experiment on **DreamerV3 + RSSM-OP**. As shown in Table 7, incorporating RSSM-OP consistently improves the performance of DreamerV3, indicating that observation prediction alone provides measurable gains. However, the highest performance is achieved when observation prediction and event prediction are learned jointly. This demonstrates that the performance improvement of EAWM mainly stems from the joint modeling of observations and events, not from RSSM-OP alone. We have included this results in the Table 13 of our manuscript and revised the description.
> $$ \\textbf{Table 7. Game scores and human normalized mean score for DreamerV3 with the RSSM-OP on the 6 games from the Atari 100K.}$$
> | Game                       | EADream         | DreamerV3 + RSSM-OP | DreamerV3       |
> |:--------------------------:|:---------------:|:-----------------:|:---------------:|
> | Assault                    | **883.4**  | 792.5             | 723.6           |
> | Breakout                   | **71.8**   | 20.8              | 26.9            |
> | Gopher                     | **4415.8** | 5287.3            | 4074.9          |
> | Krull                      | **8729.6**          | 7612.1            | 7796.6          |
> | Ms Pacman                  | 1580.7          | 1429.3            | **1813.3** |
> | Up N Down                  | **28408.2**         | 20696.9        | 20183.2         |
> | **Mean** ($\\uparrow$) | **2.501**  | 1.979 | 1.901           |
> > **Weakness 7:** (Minor) Missing citations to Craftax and DMC in Section 4.1.
>
> Thank you for the correction. We have added citations to Craftax (Matthews et al., 2024) and DMC (Tunyasuvunakool et al., 2020) in Section 4.1.
>
> > **Question 1:** The method adds several hyperparameters on top of already hyperparameter-heavy world models. How sensitive are the new hyperparameters in practice, and how difficult was tuning?
>
> **Response:**
> In terms of practical effort, tuning the additional hyperparameters required only a small number of validation runs on 6 Atari games that we used in our ablation studies: *Assault*, *Breakout*, *Gopher*, *Krull*, and *Ms Pacman*, *Up N Down*. Compared to previous work, such as DreamerV3, which tunes the base world model across many hyperparameter settings, this cost is relatively minor. In our experiment, once a reasonable default for the hyperparameters is chosen on these 6 Atari games, the values of hyperparameters transfer well to tasks across DMC, Craftax, and DMC-GB2.
>
> To analyze the sensitivity of the hyperparameters introduced by EAWM, we conducted additional experiments on these six Atari games during the rebuttal period, focusing on the following parameters:
> - The AGMM event-detection threshold, $C\_I \\in \\{8, 16, 32\\}$ (see Table 5 in the response to Reviewer z3Wd).
> - The weight on the event-related loss terms in Equation (8), $\\omega \\in \\{0, 0.5, 1\\}$ (see Table 6 in the response to Weakness 6).
>
> We posit that a universally applicable configuration exists across diverse scenarios. Accordingly, we adopt the default values $C\_I = 16$ and $\\omega = 0.5$ for all experiments across benchmarks. Notably, EADream consistently outperforms DreamerV3 even when these hyperparameters vary significantly. To clarify this point, we have included a brief discussion of hyperparameter sensitivity and our tuning protocol in Appendix F of the revised manuscript.

---

> ### Author Response · Authors · 2025-11-26
> **Rebuttal by Authors**
>
> > **Question 2:** Please address the clarification requests in the weaknesses above.
>
> **Response:** Kindly refer to the responses above. All modifications in the manuscript have been highlighted $\\color{blue}{\\text{in blue}}$. For a clearer understanding of our framework, we have shared visualizations of EAWM’s predictions on the  [anonymous GitHub repository](https://anonymous.4open.science/r/EAWM/README.md) for your convenience.
>
> ---
> **References**
>
> Zoran Zivkovic and Ferdinand Van Der Heijden. Efficient adaptive density estimation per image pixel for the task of background subtraction. Pattern recognition letters, 27(7):773–780, 2006
>
> Dylan Richard Muir and Sadique Sheik. The road to commercial success for neuromorphic technologies. Nature Communications, 16(1):3586, 2025.
>
> Michael Matthews, Michael Beukman, Benjamin Ellis, Mikayel Samvelyan, Matthew Jackson, Samuel Coward, and Jakob Foerster. Craftax: A lightning-fast benchmark for open-ended reinforcement learning. In the International Conference on Machine Learning (ICML), 2024.
>
> Saran Tunyasuvunakool, Alistair Muldal, Yotam Doron, Siqi Liu, Steven Bohez, Josh Merel, Tom Erez, Timothy Lillicrap, Nicolas Heess, and Yuval Tassa. Dm control: Software and tasks for continuous control. Software Impacts, 6:100022, 2020.
>
> Danijar Hafner, Jurgis Pasukonis, Jimmy Ba, and Timothy Lillicrap. Mastering diverse domains through world models. arXiv preprint arXiv:2301.04104, 2023.
>
> Lior Cohen, Kaixin Wang, Bingyi Kang, Uri Gadot, and Shie Mannor. Uncovering untapped potential in sample-efficient world model agents. arXiv preprint arXiv:2502.11537, 2025.

---

### Official Review · Reviewer_z3Wd · 2025-10-31

**Soundness:** 3
**Presentation:** 3
**Contribution:** 3
**Rating:** 8
**Confidence:** 3

**Summary:**

The paper suggest to extend a typically world model for reinforcement learning (e.g. Dreamer) by an event predictor.  The paper claims that that addition leads to better latent states.  Experiments show that those latent states indeed achieve better performance on various benchmarks.

**Strengths:**

- well written introduction

- excellent results across different benchmark datasets, e.g. Atari 100K

- the framework can be hooked to existing methods, e.g, Simulus and Dreamer, and consistently improves results

- the additional events are auto-labeled using an "Automated Event Generator" (based on Adaptive GMMs)

**Weaknesses:**

- not much to criticize!

**Questions:**

- Eq. (1): $p_i$ doesn't appear in the formula!  when is $p_i$ positive or negative for visual inputs?

- How sensitive is the method to the choice of $C_I$ in Eq. (1)?

- Automated Event Generator:  what are events e.g., in the game of Pong?

- I didn't really understand, what is the point of the "Generic Event Segmentator"?  What do you mean with "reallocate attention from events to raw observations"?

---

> ### Author Response · Authors · 2025-11-26
> **Rebuttal by Authors**
>
> We are sincerely grateful to the reviewer for the positive assessment of our manuscript and the feedback on presentation and ablation study. We have addressed all the raised points in detail below.
>
> ---
>
> > **Question 1:** Eq. (1): $p_i$ doesn't appear in the formula! When is $p_i$ positive or negative for visual inputs?
>
> **Response:** Thank you for the close reading. We have rewritten Eq. (1) to clarify the definition of $p_i$:
>
> $$p_i =\\left\\{ \\begin{array}{ll} +1, &\\text{if } L_t(x_i,y_i) - L_{t-1}(x_i,y_i) > C_I \\\\
> -1, &\\text{if } L_t(x_i,y_i) - L_{t-1}(x_i,y_i)< -C_I
> \\\\ 0, &\\text{otherwise} \\end{array}\\right.
> $$
> > **Question 2:** How sensitive is the method to the choice of $C_I$ in Eq. (1)?
>
> **Response:** That's a great point.  Consistent with methodologies in event-based vision (Chakravarthi et al., 2024), we posit that a universally applicable configuration exists for diverse scenarios. We use the default value  $C_I=16$ across all the benchmarks. We have conducted an additional study on the sensitivity of $C_I$ with the proposed EADream with 2 more values on 6 games from Atari 100K. We have included the sensitivity analysis in Appendix F of our revised manuscript.
> $$ \\textbf{Table 5: Game scores and human normalized mean score for different values of $C_I$ on the 6 games from the Atari 100K. } $$
> |Game|EADream $(C_I=8)$|EADream $(C_I=16)$|EADream $(C_I=32)$|DreamerV3|
> |--|--|--|--|--|
> |Assault|781.5|**883.4**|821.1|723.6
> |Breakout|43.2|**71.8**|23.1|26.9
> |Gopher|2997.1|**4415.8**|3316.5|4074.9
> |Krull|7043.0|**8729.6**|7575.8|7796.6
> |Ms Pacman|1568.6|1580.7|1566.5|**1813.3**
> |Up N Down|38616.1|28408.2|**42507.2**|20183.2
> |**Mean** ($\\uparrow$)|2.082|**2.501**|2.144|1.901
>
> > **Question 3:** Automated Event Generator: What are events, e.g., in the game of Pong?
>
> **Response:** For visual inputs, instead of using the primary definition of events, we generate events using AGMM. In the game of Pong, [typical events](https://anonymous.4open.science/r/EAWM/visualization/atari/event/pong.gif) include
>   * Ball–paddle contact,
>   * Score updates at the top
>   * Ball position change and paddle motion.
>
> At the [Anonymous GitHub](https://anonymous.4open.science/r/EAWM/README.md), we have visuallized the events generated by the Automated Event Generator on the Atari, DMC, and DMC-GB2.
>
> > **Question 4:** I didn't really understand. What is the point of the "Generic Event Segmentator"? What do you mean by "reallocate attention from events to raw observations"?
>
> **Response:** We apologize for the unclear description of the Generic Event Segmentator (GES) in the original paper. The GES is designed to identify event boundaries, *i.e.*, the start or end time of a meaningful slice of an observation stream (Zacks & Swallow, 2007). With "reallocate attention from events to raw observations", we mean that world models should temporarily stop predicting events and instead focus on modeling raw observations. We have clarified this point, and here's a summary of our revised manuscript:
>
> From the perspective of cognitive science, humans segment continuous sensory streams into discrete events for decision-making (Soon et al., 2022). Human comprehension systems tend to form future predictions within event boundaries, whereas humans' ability to predict and memory decreases at the event boundaries (Radvansky & Zacks, 2017). Motivated by this, we develop a generic event segmentor (GES) that automatically detects event boundaries. If there is no event boundary, world models should prioritize dynamic events over static observations so as to improve accuracy on informative parts of the observations, rather than uniformly over all observations. Otherwise, when the GES outputs 0 and an event boundary is detected, world models should stop predicting events and instead focus on modeling raw observations. In other words, the world model should reallocate attention from events to raw observations at the event boundary, where event prediction is devoid of meaning. We implement this via an event-aware observation loss:
> $$
>     \\mathcal{L}\_{\\text{o}}(\\theta)\\doteq \\mathcal{L}'\_{\\text{o}}(\\mathbf{o}\_{t},\\hat{\\mathbf{o}}\_{t})+
>     \\sum\_{m=1}^{M}\\sum\_{i=1}^{N^m}\\omega g(\\alpha^{(m)}\_t,\\alpha\_{\\text{thr}}^{(m)})\\left[\\mathbb{I}(e\_{t,i}^{(m)}  \\text{occurs})-1\\right]\\mathcal{L}'\_{\\text{o}}(o\_{t,i},\\hat{o}\_{t,i}),
> $$
> where $\\omega \\in [0,1]$ is the event-aware weight to balance attention of the overall observations and the part of the observations where events take place. $\\mathcal{L}'\_{\\text{o}}$ denotes the loss function of observations given by base world models by default if it exists.

---

> > ### Author Response · Authors · 2025-11-26
> > **Rebuttal by Authors**
> >
> > ---
> > **References**
> >
> > Shin, Yeon Soon, and Sarah DuBrow. Structuring memory through inference‐based event segmentation. Topics in cognitive science, 13(1), pp.106-127, 2022.
> >
> > Bharatesh Chakravarthi, Aayush Atul Verma, Kostas Daniilidis, Cornelia Fermuller, and Yezhou Yang. Recent event camera innovations: A survey. In European Conference on Computer Vision, pp. 342–376. Springer, 2024.
> >
> > Jeffrey M Zacks and Khena M Swallow. Event segmentation. Current directions in psychological science, 16(2):80–84, 2007.
> >
> > Gabriel A Radvansky and Jeffrey M Zacks. Event boundaries in memory and cognition. Current opinion in behavioral sciences, 17:133–140, 2017

---

### Official Review · Reviewer_zrEf · 2025-11-01

**Soundness:** 3
**Presentation:** 3
**Contribution:** 2
**Rating:** 4
**Confidence:** 5

**Summary:**

This paper proposes Event-Aware World Model (EAWM), a framework that enhances model-based reinforcement learning by predicting events rather than raw observations. The method introduces three key components: an automated event generator that extracts events from multi-modal observations without manual labels, an event predictor that shapes representations through information bottleneck optimization, and a Generic Event Segmentor (GES) that identifies event boundaries to stabilize training.

**Strengths:**

* Well-motivated approach.
* The paper demonstrates consistent and substantial improvements across diverse benchmarks.
* The paper successfully demonstrates applicability across different architectures.

**Weaknesses:**

* The core idea of integrating dynamics between frames as an additional learning signal for world models has already been explored by DyMoDreamer [1]. Moreover, DyMoDreamer achieves better performance than EADreamer on both Atari 100K and DeepMind Control Vision benchmarks with a simpler method.
* While the authors claim to learn task-relevant environmental dynamics, the EAWM primarily relies on inter-frame pixel differences. This approach may still filter in many task-irrelevant events.
* Adding the event detector likely increases the training time for world models. The paper does not report wall-clock training time comparisons or computational cost analysis.

[1] Dymodreamer: World Modeling with Dynamic Modulation

**Questions:**

* Why do EASimulus and EADream have different event predictor inputs? EASimulus conditions on $(y_{t-1}, y_t)$ while EADream conditions on $(h_t, \hat{z}_t, z_t)$.
* Can the authors provide qualitative imagination results on DMC-GB2? Given that DMC-GB2 tests generalization to visually noisy observations (randomized colors and video backgrounds), it would be valuable to visualize the world model's imagined trajectories and corresponding event predictions. This would help verify whether EAWM truly learns to attend to task-relevant objects while filtering out background distractors, or if the performance gains come from other factors.
* What is the wall-clock time comparison with baselines?

I am willing to raise my score if the authors can address these concerns.

---

> ### Author Response · Authors · 2025-11-26
> **Rebuttal by Authors**
>
> We thank the reviewer for the valuable feedback. We have prepared a point-by-point response to the comments below.
>
> ---
> > **Weakness 1:** The core idea of integrating dynamics between frames as an additional learning signal for world models has already been explored by DyMoDreamer. Moreover, DyMoDreamer achieves better performance than EADreamer on both Atari 100K and DeepMind Control Vision benchmarks with a simpler method.
>
> **Response:**
> Thank you for your kind reminder regarding this recent work, which has been accepted to NeurIPS 2025. In fact, DyMoDreamer was published on [ArXiv](https://arxiv.org/abs/2509.24804) on September 29, after the full paper deadline for ICLR 2026 (September 24, 2025). As stated in the [ICLR 2026 Reviewer Guide](https://iclr.cc/Conferences/2026/ReviewerGuide), comparisons with concurrent work published within the two months leading up to the full paper deadline (on or after July 24) are not required. Nevertheless, we agree that DyMoDreamer is highly relevant, and we appreciate the opportunity to discuss the relationship between the two approaches.
>
> We have cited DyMoDreamer and integrated it into the revised related work section. Besides, we have conducted a theoretical and empirical comparison between DyMoDreamer and the proposed EAWM (including EADream as an instance) as follows.
>
> - *Theoretical comparison*.
>
>    DyMoDreamer and EAWM share the motivation of enriching world models with dynamic information, but they are realized with distinct design principles and formulations.
>
>    - Distinct design principles. DyMoDreamer introduces dynamic information as input, while EAWM employs this dynamic information as a supervision signal for world model optimization, inspiring the world model to focus on kinetic features. As a result, EAWM does not introduce any additional inference-time computations, whereas DyMoDreamer requires extra processing for dynamic information.
>
>    - Different formulations. DyMoDreamer adopts pixel-level difference between adjacent frames as the dynamic information, while EAWM captures discrete kinetic events (*e.g.*, changes in brightness, proprioceptive signals, or token switches) with identified start/end time (the event boundaries).
>
>     - Visual generalization. While DyMoDreamer ignores static background noise, its generalization ability remains vulnerable to low-level visual variations, such as illumination or background changes. In contrast, EAWM learns robust event-aware representations that abstract away from raw pixel fluctuations and capture semantically meaningful events, which offers the promise of transferability in unseen environments with complex visual distractors and background changes.
>
>    The concrete implementations of the two models are described below.
>
>    DyMoDreamer injects masked images as inputs inside a DreamerV3 architecture, focusing primarily on dynamics for visual inputs. Specifically, given a predefined threshold $\\epsilon$, DyMoDreamer defines a mask $\\textbf{M}\_t$ for the current observation $\\mathbf{o}\_t$, where each element $M\_{t,h,w,c}=1$ if $\\Vert o\_{t,h,w,c}- o\_{t-1,h,w,c}\\Vert\_2 > \\epsilon.$ Then, masked images $\\mathbf{o}'\_t$ are generated via $\\mathbf{M}\_t\\odot \\mathbf{o}\_t$. Then, they use a dynamic encoder $ q\_{\\mathbf{\\theta}}(\\mathbf{d}\_t |\\mathbf{h}\_t, \\textbf{o}'\_t)$ to process $\\mathbf{o}'\_t$ to generate dynamic modulators $\\mathbf{d}\_t$, which are subsequently incorporated into the sequence model $\\mathbf{h}\_{t}= \\mathbf{F}\_{\\theta}(\\mathbf{h}\_{t-1},\\mathbf{z}\_{t-1},\\mathbf{d}\_{t-1}, \\mathbf{a}\_{t-1})$. While effective for pixel-based representations, this design limits the practicality of DyMoDreamer for token-based world models such as Simulus, as it introduces an additional stream of tokens of dynamic modulators $\\mathbf{d}\_t$, thereby increasing computational overhead by a large margin.
>
>    Instead of treating inter-frame differences as pixel-level inputs, EAWM formulates events as the fundamental supervision signal. We introduce the automated event generator that converts raw multi-modal observations into discrete kinetic events and the Generic Event Segmentor (GES) that identifies event boundaries to obtain semantically meaningful segments. EAWM is trained to predict both future observations and events via Equation (9) in our manuscript:
>    $$ \\mathcal{L}\_(\\theta)\\doteq\\mathcal{L}\_{\\text{WM}}(\\theta)+\\mathcal{L}\_{\\text{EA}}(\\theta)=\\mathcal{L}\_{\\text{WM}}(\\theta)+\\beta\_o\\mathcal{L}\_o(\\theta)+\\beta\_e\\mathcal{L}\_e(\\theta), $$
>    where $\\mathcal{L}\_{\\text{WM}}(\\theta)$ denotes the loss functions of the base world model, except the loss for observations. This allows us not only to preserve the benefits of dynamic modulation seen in DyMoDreamer, but also to learn event-aware representations as a broadly applicable principle for accelerating policy learning across diverse architectures and environments.

---

> ### Author Response · Authors · 2025-11-26
> **Rebuttal by Authors**
>
> - *Empirical comparison.*
>
> In practice, we conducted experiments to compare the
>    - **Architectural flexibility**. Based on DreamerV3, DyMoDreamer demonstrated its effectiveness on 3 benchmarks. In comparison, Figure 2 in the original manuscript demonstrates that EAWM can be attached to a wide family of existing world models with minimal modifications via the instantiation of the EADream and EASimulus across 4 established benchmarks.
>
>    - **Computational cost**. On a V100 GPU, training an EAWM model cost us 2.6 days on average for a task from DMC-GB2, while training a DyMoDreamer model cost us 3.6 days.
>    - **Visual generalization**. To compare the visual generalization of EADream and DymoDreamer, we have implemented DyMoDreamer on DMC-GB2. Table 1 calculates the average of scores over 3 seeds in *Color Hard*, *Video Hard*, and *Color Video Hard* test environments. Tables 2-4 showcase the detailed scores for each task in test environments. DyMoDreamer achieves a mean score of 448 and performs even worse than DreamerV3 in *Video Hard* and *Color Video Hard* test environments, where the background is substituted for video from natural environments, since they rely heavily on the frame differencing method. Considering that our EADream obtained the mean score of 606, we believe that EAWM shows stronger potential than DyMoDreamer for transferring to broader environments where visual distractors are pervasive.
>
> $$ \\textbf{Table 1: Average Scores achieved in hard test sets of DMC-GB2} $$
> |Test Environments|DreamerV3|DyMoDreamer|EADream (Ours)|
> |--|:--:|:--:|:--:|
> |Color Hard|456| $\\underline{\\text{613}}$|**750**|
> |Video Hard|$\\underline{\\text{343}}$|339|**477**|
> |Color Video Hard|$\\underline{\\text{402}}$|391|**590**|
> |Average Score|419 |$\\underline{\\text{448}}$|**606**|
>
> $$ \\textbf{Table 2: Detailed scores achieved in Color Hard test environments} $$
> |Task|DreamerV3|DyMoDreamer|EADream (Ours)|
> |--|:--:|:--:|:--:|
> |Cartpole Swingup|432|$\\underline{\\text{495}}$|**774**|
> |Cheetah Run|417|**593**|$\\underline{\\text{567}}$|
> |Cup Catch|93|$\\underline{\\text{175}}$|**914**|
> |Finger Spin|218|**681**|$\\underline{\\text{576}}$|
> |Walker Stand|957|**969**|$\\underline{\\text{964}}$|
> |Walker Walk|617|**766**|$\\underline{\\text{705}}$|
> |**Mean**($\uparrow$)|456| $\\underline{\\text{613}}$|**750**|
>
> $$ \\textbf{Table 3: Detailed scores achieved in Video Hard test environments} $$
> |Task|DreamerV3|DyMoDreamer|EADream (Ours)|
> |--|:--:|:--:|:--:|
> |Cartpole Swingup|$\\underline{\\text{296}}$|138|**449**|
> |Cheetah Run|$\\underline{\\text{228}}$|127|**240**|
> |Cup Catch|53|$\\underline{\\text{92}}$|**288**|
> |Finger Spin|99|$\\underline{\\text{205}}$|**400**|
> |Walker Stand|$\\underline{\\text{816}}$|809|**872**|
> |Walker Walk|569|**664**|$\\underline{\\text{613}}$|
> |**Mean**($\uparrow$)|$\\underline{\\text{343}}$|339|**477**|
>
> $$ \\textbf{Table 4: Detailed scores achieved in Color Video Hard test environments} $$
> |Task|DreamerV3|DyMoDreamer|EADream (Ours)|
> |--|:--:|:--:|:--:|
> |Cartpole Swingup|$\\underline{\\text{338}}$|137|**437**|
> |Cheetah Run|$\\underline{\\text{435}}$|356|**531**|
> |Cup Catch|16|$\\underline{\\text{87}}$|**573**|
> |Finger Spin|63|$\\underline{\\text{149}}$|**398**|
> |Walker Stand|$\\underline{\\text{946}}$|837|**952**|
> |Walker Walk|614|**778**|$\\underline{\\text{648}}$|
> |**Mean**($\\uparrow$)|$\\underline{\\text{402}}$|391|**590**|

---

> ### Author Response · Authors · 2025-11-26
> **Rebuttal by Authors**
>
> > **Weakness 2:** While the authors claim to learn task-relevant environmental dynamics, the EAWM primarily relies on inter-frame pixel differences. This approach may still filter in many task-irrelevant events.
>
> **Response:**
> We appreciate the reviewer’s concern and agree that naive inter-frame differencing would indeed admit many task-irrelevant changes. However, EAWM’s event generator and GES are more structured than simple pixel differencing and are explicitly designed to suppress many nuisance variations.
>
> 1. **Beyond raw inter-frame differences.**
>    Although the events are derived from video frames, they are *not* triggered by simple pixel-wise differences. Instead, for each pixel $(x\_i, y\_i)$, we employ an Adaptive Gaussian Mixture Model (AGMM) over luminance $L\_{t+1}(x\_i,y\_i)$: We compute the squared Mahalanobis distance $D\_{k,i}$ between $L\_{t+1}(x\_i,y\_i)$ and the component means. An event is only generated when the observation is a *statistically significant deviation* from this learned multi-modal distribution of the background (large $D\_{k,i}$) or when the matched component has low weight relative to the others (high uncertainty).   many task-irrelevant fluctuations—such as small sensor noise or slow global illumination changes—are absorbed by the evolving mixture components and do *not* trigger events. It is worth noting that the events generated by AGMMs for visual inputs serve only as the spatial attention priors for world model training.
>
> 2. **Suppression of task-irrelevant events for downstream control.**
>    While the event-aware representation learning focuses the model’s capacity on informative changes, task relevance for downstream control is induced by the rewards and continuation flags. Even if some individual events generated by AGMM are not directly task-critical, our GES can automatically detect event boundaries where the prediction of such events is devoid of meaning. We assume that task-relevant events and task-irrelevant changes do not always co-occur. Thus, the co-learning objectives for EAWM (e.g., rewards and events) induce a biased neural state: it enhances understanding of tasks, whilst suppressing the neural response to irrelevant information. Since the predictions of world models are not perfect, most irrelevant events generated by AGMMs may be hard to capture due to the information bottleneck. As a result, EAWM pays more attention to *meaningful spatio-temporal transitions*, and yields compact kinetic features that accelerate policy learning in general.
>
>    Formally, the world model parameters $\\theta$ are optimized using collected data by the agent whose goal is to complete the task, which selectively emphasizes the task-relevant environmental dynamics. As shown in Equation (7), the general loss function of event prediction is controlled by the GES.
>    Furthermore, world models improve accuracy on informative parts of the observations, rather than uniformly over all observations. However, when GES detects an event boundary where event prediction is devoid of meaning, the world model should reallocate attention from events to raw observations. We implement this via an event-aware observation loss:
>    $$\\mathcal{L}\_{\\text{o}}(\\theta)\\doteq \\mathcal{L}'\_{\\text{o}}(\\mathbf{o}\_{t},\\hat{\\mathbf{o}}\_{t})+
>     \\sum\_{m=1}^{M}\\sum\_{i=1}^{N^m}\\omega g(\\alpha^{(m)}\_t,\\alpha\_{\\text{thr}}^{(m)})\\left[\\mathbb{I}(e\_{t,i}^{(m)}  \\text{ occurs})-1\\right]\\mathcal{L}'\_{\\text{o}}(o\_{t,i},\\hat{o}\_{t,i}).$$
>
>    The overall effect of training is to bias representation learning toward parts of the dynamics where **task-relevant events** tend to occur. Noisy events that do not aid in predicting future value are naturally suppressed during the training of the world models.
>
> 3. **Practical behavior.**
>    Please refer to our visualization at the [Anonymous GitHub](https://anonymous.4open.science/r/EAWM/README.md). We observe that:
>
>    - The event stream is **highly sparse** compared to dense frame-wise processing, indicating that many small and irrelevant changes are discarded.
>    - The events tend to cluster around agent actions and salient environment changes (collisions, object appearances/disappearances, etc.), which are precisely the types of dynamics that the policy network needs to receive.
>    - The capability of capturing task-relevant events by EAWM generalizes well to unseen test environments of DMC-GB2, which can be credited to the event-aware representation learning.
>
> In the revised manuscript, we have clarified this point by emphasizing that our event generator for visual inputs uses statistical novelty and uncertainty from the stream of observations via AGMM, rather than the primary definition of events in Equation (1), and provided a deeper analysis of our GES. In addition, we have included a short qualitative illustration of the kinds of events that are retained in typical tasks, as shown in Appendix N.2.

---

> > ### Comment · Reviewer_zrEf · 2025-11-27
> >
> > Thank you for the comprehensive rebuttal. The authors' responses effectively address my concerns, based on these responses, I am raising my score from 4 to 6.

---

> > > ### Author Response · Authors · 2025-11-27
> > >
> > > Dear reviewer, we are truly grateful for the positive reconsideration of our work and your constructive feedback, which has played a crucial role in enhancing the overall quality of the paper. We remain committed to further strengthening the manuscript wherever possible.

---

> ### Author Response · Authors · 2025-11-26
> **Rebuttal by Authors**
>
> > **Weakness 3:** Adding the event detector likely increases the training time for world models. The paper does not report wall-clock training time comparisons or computational cost analysis.
>
> **Response:**
>
> Thanks for your practical suggestion. We report the average wall time of per run on each benchmark, including the time cost of the automated labeling program into account.
>
> Overall, our methods incur only marginal additional training costs relative to conventional world model optimization since the event detector introduces negligible GPU memory cost. We have included the wall-clock training time and other metrics for computational complexities in Appendix L of the revised manuscript.
>
> **EADream** Our experiments of EADream on Atari 100K were conducted with NVIDIA V100 32GB GPUs. Training on Atari 100K, with three tasks running on the same GPU in parallel, took about 1.2 days, resulting in an average of 0.4 days per environment (+11% more than DreamerV3). Experiments on DMC were conducted with NVIDIA GeForce RTX 4090 24GB GPUs. Training on DMC, with three
> tasks running on the same GPU in parallel, took 1.8 days, resulting in an average of 0.6 days per
> environment (+13% more than DreamerV3). On a V100 GPU, training a baseline DreamerV3 on DMC-GB2 cost us 2.3 days, while training an EADream model cost us 2.6 days.
>
> **Simulus** Our experimental evaluations of Simulus were finished on NVIDIA GeForce RTX 4090 24GB GPUs. Training a baseline Simulus model on Atari 100K and Craftax 1M required approximately 0.7 day and 6 days per task, respectively. Compared to the training cost of our EASimulus
> (on Atari 100K and Craftax 1M, approximately 0.8 day and 7 days per task, respectively).
>
> > **Question 1:** Why do EASimulus and EADream have different event predictor inputs?
>
> **Response:** Thanks for your close reading. This is not an inconsistency in the framework, but a consequence of how EAWM is instantiated on two structurally different base world models, under a minimal-modification design principle. In Equation (3), events are predicted from the baseline world model’s predictive state. For DreamerV3, this corresponds naturally to ($\\mathbf{h}\_t,\\hat{\\mathbf{z}}\_t,\\mathbf{z}\_t$), where $\\mathbf{h}\_t=\\mathbf{y}\_t$ is the recurrent state and $\\hat{\\mathbf{z}}\_t/\\mathbf{z}\_t$ are prior/posterior latents. The future observations, rewards, and continuation flags are predicted from the triplet, and hence conditioning on this triplet directly provides enough information to capture events. Simulus, in contrast, explicitly decouples the encoder–decoder network from the Retentive Network sequence model, and uses $\\mathbf{y}\_t$ as the segment-level predictive representation. In other words, the future observations, rewards, and continuation flags are computed via $\\mathbf{y}\_t$ under their framework of Simulus. Thus, attaching the event predictor to ($\\mathbf{y}\_{t-1},\\mathbf{y}\_{t}$) lets us capture events while preserving this modular training setup without feeding gradients back through the vision encoder. Therefore, although the parameterizations differ, both EADream and EASimulus follow a unified principle: events are consistently inferred from the native predictive state of their respective base world models. This ensures architectural compatibility, maintains computational efficiency, and allows EAWM to capture meaningful spatio-temporal transitions via information-bottleneck-driven optimization.
>
>
> > **Question 2:** Can the authors provide qualitative imagination results on DMC-GB2?
>
> **Response:**
> This is a great point. Although the automated event generator may introduce some irrelevant events, our event predictor successfully filters out such visual distractors within EAWM's imagined trajectories. We have put the visualization of imagined observations and predicted events in both train and test environments from DMC-GB2 at [the Anonymous GitHub](https://anonymous.4open.science/r/EAWM/README.md). The visualization of event predictions proves that EAWM is capable of capturing task-relevant events and filtering out visual distractors. We observe that event prediction generalizes well to unseen test environments where the background is changing continuously strong potential for transfer to broader environments. These results provide a reasonable rationale for the excellent performance of our EAWM on DMC-GB2.
>
> Thanks for your constructive suggestion that deepens our understanding of the EAWM. We have presented qualitative imagination results and analysis on DMC-GB2 in Appendix N.2 of our revised manuscript.
>
>
> > **Question 3:** What is the wall-clock time comparison with baselines?
>
> **Response:**
>  Please refer to our response to Weakness 3.

---

### Meta-Review · Area_Chair_NqKf · 2026-01-04

**Summary:**

This paper introduces the Event-Aware World Model , which integrates event-driven characteristics into the training of world models. By learning event-aware representations, the model prioritizes dynamic information in the input while mitigating the impact of task-irrelevant noise. The framework utilizes an automated event generator for automated event labeling and incorporates a Generic Event Segmentor to facilitate learning. The authors demonstrate the effectiveness of this approach through strong performance across multiple benchmarks.

**Reviewer Concerns:**

- Reviewer zref: The primary concern was the technical similarity and differences between the proposed EAWM and DymoDreamer. The authors addressed this by providing a detailed theoretical analysis and additional experimental comparisons to demonstrate EAWM's specific advantages.

- Reviewer z3wd: This reviewer expressed confusion regarding certain concepts and the overall clarity of expression in the original manuscript. The authors provided comprehensive explanations to clarify these points.

- Reviewer 6jd5: This reviewer concerned about the presentation and writing quality of the paper. The authors have since revised the manuscript to improve readability and flow according to the reviewer's feedback.

The authors have effectively addressed the major concerns raised during the discussion period. The distinction from prior work has been clarified, and the writing quality has been significantly improved.

**Reviewer Scores:**

- Reviewer zref: already mentioned in his comment that he will raised the score to 6.

- Reviewer z3wd: would maintain the current score following the clarifications.

- Reviewer 6jd5: Explicitly stated he would raise their score to Acceptance if his concerns were addressed. Given the extensive revisions made by the authors, I believe this increase is justified and highly likely.

---

### Decision · Program_Chairs · 2026-01-26

Accept (Poster)